

# Mitigating adversarial attacks in federated learning based network traffic classification applications using secure hierarchical remote attestation and adaptive aggregation framework

Azizi Ariffin[1,2], Faiz Zaki[1], Hazim Hanif[1,3] and Nor Badrul Anuar[1,4]

[1] Centre of Research for Cyber Security and Network (CSNET), Faculty of Computer Science and Information Technology, Universiti Malaya, Kuala Lumpur, Malaysia
[2] School of Computing Sciences, College of Computing, Informatics and Mathematics, Universiti Teknologi MARA, Shah Alam, Selangor, Malaysia
[3] Department of Software Engineering, Faculty of Computer Science and Information Technology, Universiti Malaya, Kuala Lumpur, Malaysia
[4] Institute of Informatics & Computing in Energy, Universiti Tenaga Nasional, Kajang, Selangor, Malaysia

## ABSTRACT

**Background:** Federated learning (FL) enhances network traffic classification (NTC) with significant benefits in privacy, performance, and efficiency. However, the distributed nature of FL exposes NTC models to critical adversarial attacks from byzantine clients. These attacks, such as label flipping (LF) and model poisoning, can severely degrade overall model performance, while backdoor and generative adversarial network (GAN) based attacks can force the model to misclassify specific traffic classes. Securing FL-based NTC is paramount, as these vulnerabilities pose substantial threats to its vital role in network management, quality of service, and threat identification.

**Methods:** While various defensive measures for FL exist, they are often ineffective against multiple types of adversarial attacks, and their effectiveness diminishes as the number of attackers increases. To address this gap, this study proposed SHeRAA-FL, a secure framework for FL-based NTC. The framework secures the training process by combining remote attestation scoring, hierarchical training, and adaptive aggregation mechanisms, reinforced with hardware-level security and encrypted communication. We developed and evaluated SHeRAA-FL on public datasets, such as ISCX-VPN 2016 and N-BaIoT, benchmarking it against existing approaches, including weighted averaging, median-mean, trim-mean, Krum, and Multi-Krum.

**Results:** The evaluation results show that SHeRAA-FL effectively mitigates the impact of multiple types of adversarial attacks, even in scenarios with multiple attackers. For example, in the LF attack, other approaches recorded a 99.6% accuracy reduction, while SHeRAA-FL only recorded a 5.33% reduction. Moreover, in a normal scenario, the framework produces a model with the highest accuracy of 0.9130, indicating minimal disruption to the FL process.

Corresponding authors
Faiz Zaki, faizzaki@um.edu.my
Nor Badrul Anuar,
badrul@um.edu.my

## INTRODUCTION

Network traffic classification (NTC) categorizes network data into predefined classes (*e.g.*, Facebook, YouTube, Productivity/Gaming, Malicious/Benign). NTC enables effective network management by leveraging on the increased visibility to perform tasks, such as quality of service (QoS) provisioning, device fingerprinting, and threat identification (*Zaki et al., 2021*). Applying federated learning (FL) to NTC provides numerous benefits. This machine learning approach enables multiple clients to collaboratively train a model by sharing only model updates, thus keeping their raw data private. Primarily, FL applications enhance data privacy when training NTC models on large volumes of traffic data, especially with deep learning algorithms (*Guo & Wang, 2023*). This enhanced privacy encourages organizations to share model parameters as a form of collaborative intelligence. This intelligence sharing helps the NTC model generalize across various network environments and achieve higher classification performance (*de Carvalho Bertoli et al., 2023*; *Sarhan et al., 2022*). Furthermore, FL enables model training on edge devices, which reduces latency and bandwidth utilization (*Kukreja et al., 2019*). This deployment proximity to user endpoints improves the classifier's overall responsiveness.

However, the distributed nature of FL training makes FL-based NTC vulnerable to various adversarial attacks, including label flipping (LF), model poisoning (MP), backdoor, and generative adversarial network (GAN) based attacks. This vulnerability exists because any participating FL client can become a Byzantine node and sabotage the training process, either intentionally or unintentionally. A malicious Byzantine node can sabotage FL training by conducting attacks that either degrade the NTC model's performance (*Rey et al., 2022*) or make the model produce specific outcomes that favour the attacker (*Holodnak et al., 2022*). This poses significant security implications, as NTC plays a vital role in network management, QoS, and threat identification. For example, the average cost of a security breach reached USD 4.88 million in 2024, placing a significant financial burden on organizations (*IBM, 2024*). Moreover, these security issues erode trust in FL-based NTC, discouraging organizations from participating in the collaborative training needed to improve model generalization.

To address these security issues, researchers have introduced various defensive strategies for FL training, including value-based (*Cao & Gong, 2022*), distance-based (*Cao et al., 2024*), and weighted-averaging approaches (*Zhou et al., 2022*). However, several limitations in these approaches reduce their effectiveness at mitigating multiple types of adversarial attacks.

For example, the effectiveness of value-based approaches like trimmed-mean decreases as the number of attackers increases, and these methods only work against certain types of attacks (*Rey et al., 2022*). Similarly, distance-based approaches such as Krum work under the assumption that malicious clients constitute less than 33% of the total clients

(*Blanchard et al., 2017*). Finally, the effectiveness of the weighted-averaging approach depends on its ability to assign weights to benign clients correctly; this method only reduces an attack's influence rather than preventing it entirely (*Xie et al., 2020*).

In addition, the effectiveness of approaches using anomaly detection or reference models depends on the availability of public datasets (*Raza et al., 2022*). Furthermore, sign-based approaches may fail to detect subtle attacks where the poisoned data does not alter the update's direction (*Guo, Xu & Zhu, 2023*). Analysis of previous works reveals that most existing defensive approaches are effective only against specific types of attacks, rather than multiple types. The effectiveness of these measures also diminishes as the number of attackers increases, potentially overwhelming the FL training process. Lastly, many existing measures lack crucial security features, including hardware-level security, encrypted communication, and identity verification. This absence of features exposes FL training to external tampering, privacy leakage, and identity-based exploits like Sybil attacks (*Xiao et al., 2022*).

To address this gap, this study proposes and develops a secure, hierarchical remote attestation with an adaptive aggregation federated learning framework named SHeRAA-FL. The framework contains three primary mechanisms: remote attestation scoring, hierarchical training, and adaptive aggregation. The remote attestation scoring mechanism verifies client trustworthiness, while the hierarchical training mechanism minimizes security risks by clustering clients into similar domains. The adaptive aggregation mechanism provides a dynamic method to filter and minimize the influence of multiple poisonous updates.

In addition, the framework leverages hardware-level security to prevent tampering and transport layer security (TLS) to ensure private communication between hosts. We developed the framework in Python 3 and made the source code available *via* GitHub. To evaluate the framework's effectiveness, this study simulates various adversarial attacks with multiple attackers using the ISCX-VPN 2016, Fashion-MNIST, N-BaIoT, and CIFAR-10 datasets. In summary, this article makes the following contributions:

(a) We propose SHeRAA-FL, a secure, hierarchical remote attestation with an adaptive aggregation federated learning framework that mitigates the impact of multiple adversarial attacks when training an FL-based NTC model.

(b) We developed the framework and federated learning testbed using Python3 and software libraries related to deep learning, federated learning, networking, cryptography, and the trusted platform module.

(c) We evaluate the framework against four types of adversarial attacks: LF, MP, Backdoor, and GAN-based attacks, using ISCX-VPN 2016, Fashion-MNIST, N-BaIoT and CIFAR-10 datasets. We also benchmark the results with existing defensive measures such as trim-mean, median-mean, weighted averaging, Krum, and multi-Krum.

The evaluation results demonstrate that SHeRAA-FL is highly effective at mitigating multiple types of adversarial attacks, even from multiple attackers, while introducing minimal disruption in normal scenarios. It maintains high model accuracy with only slight

reductions, such as 5.33% in LF, 1.13% in model cancelling, and 0.08% in gradient factor attacks, significantly outperforming other measures like Krum, which saw a 99.6% accuracy drop. The framework can mitigate backdoor attacks, achieving a zero success rate where other defenses allowed over 40%, and effectively defends against GAN-based and class-LF attacks by maintaining high F1-scores. Besides that, in a non-adversarial environment, SHeRAA-FL achieves the highest accuracy (0.9131) compared to other aggregation algorithms, proving its robustness and efficiency.

This study organizes the remainder of this article as follows: The second section discusses related works on FL-based NTC, adversarial attacks, and defensive measures in FL. The third section discusses the FL-based NTC architecture and training process. The fourth section describes this study's methodology, including the design of the proposed framework. The fifth section presents and discusses experimental results that evaluate the framework. Finally, the last section concludes the article and discusses future work. Table 1 lists the abbreviations, and Table 2 provides details on notations that we use in this article.

## RELATED WORKS

This section provides background on related topics, including FL-based NTC applications, adversarial attacks, and defensive approaches.

NTC classifies network traffic by analyzing patterns in protocol headers, payloads, session flows, and other network packet information. Accurate network traffic classification enables various network management applications, such as threat detection, QoS provisioning, and device fingerprinting (*Ariffin et al., 2025*). Applying FL to train deep learning (DL)-based NTC models shifts the training paradigm from centralized to distributed. This paradigm shift addresses several issues in traditional DL-based NTC training.

First, distributed training enhances data privacy because clients train the NTC model locally on their own devices using their private data (*Guo & Wang, 2023*). In FL, clients only need to send their model weight parameters to an aggregator to form a global model. Second, performing distributed training without exposing local data encourages organizations to collaborate on building more generalized NTC models (*de Carvalho Bertoli et al., 2023*; *Sarhan et al., 2022*). For example, two universities with different internet usage behaviors could share their local NTC model weights, creating a form of shared intelligence to improve their respective IDS or content filtering capabilities.

Third, a more generalized NTC model achieves higher accuracy in classifying various network services and threats (*Popoola et al., 2022*). Moreover, because FL does not require transferring large training datasets to a central entity, its distributed training approach optimizes resource utilization and reduces both latency and bandwidth consumption (*Ariffin, Zaki & Anuar, 2023*). FL's features also enable the training of DL-based NTC models on resource-constrained edge devices, such as routers, firewalls, or switches, that are closer to the user's endpoint (*Kukreja et al., 2019*). Training the NTC model closer to the user enhances the network administrator's ability to filter content effectively and protects the user from security threats. However, despite these benefits, applying FL in the NTC domain introduces security vulnerabilities that researchers must address.

**Table 1** Abbreviations used in this article.

| Abbreviation | Description |
| --- | --- |
| ASR | Attack success rate |
| DL | Deep learning |
| 1D/2D-CNN | 1/2-Dimensional convolutional neural network |
| FB | Facebook |
| FedAvg | Federated averaging |
| GAN | Generative adversarial network |
| IDS | Intrusion detection system |
| LF | Label flipping |
| MLP | Multilayer perceptron |
| MM | Median mean |
| MP | Model poisoning |
| NTC | Network traffic classification |
| PBM | Packet byte matrix |
| PBV | Packet byte vector |
| TCP/IP | Transmission control protocol/Internet protocol |
| TEE | Trusted execution environment |
| TLS | Transport layer security |
| TM | Trim mean |
| TPM | Trusted platform module |
| WA | Weighted averaging |

Distributing training tasks across various clients makes the FL process vulnerable to adversarial attacks, as any participating client can become a Byzantine node and sabotage the training. Clients can become Byzantine nodes if an external party or a malicious insider compromises the edge host. Additionally, insider attacks can occur due to client-side software malfunctions or errors. The following subsections describe common adversarial attacks against FL-based NTC:

(a) *Label flipping.* Malicious clients purposely tamper with the labels of their dataset and then train their local models using this manipulated data. When a client sends a parameter update for aggregation, it sends erroneous weights that degrade the global model's performance. In the context of traffic classification, flipping all class labels can degrade the model's overall accuracy to zero, while flipping only specific class labels primarily increases the false positive rate (*Rey et al., 2022*). Colluding clients or an attacker using a Sybil attack can further enhance the attack's impact (*Xiao et al., 2022*). In general, this attack is not subtle, as an administrator will likely notice severe drops in overall performance. Thus, *Nowroozi et al. (2024)* proposed a more subtle attack variation that degrades target class accuracy while only slightly degrading overall performance.

(b) *Model poisoning.* This attack poisons the parameter updates sent to the aggregator server by tampering with the model's weights or gradients. Consequently, when the

**Table 2 Table of notations.**

| Variable | Description | Type | Constraints/Range |
|---|---|---|---|
| **FL-based NTC architecture, secure framework design & federated learning testbed setup** | | | |
| $k$ | FL clients | Integer | $k > 1$ |
| $D_k$ | Clients' domain no | Integer | $1 \leq D_k \leq k$ |
| $N$ | Total number of samples in the dataset | Integer | $N > 1$ |
| $x_i$ | The feature vector for the i-th sample | Vector | $x_i \in \mathbb{R}^d$ |
| $\mathbb{X}_k$ | Clients' local dataset | Set | $\mathbb{X}_k = \{(x_i, y_i)\}_{i=1}^{N_k}$, where $N_k \geq 1$ |
| $\mathbb{X}_T$ | Test or evaluation dataset | Set | $\mathbb{X}_T = \{(x_i, y_i)\}_{i=1}^{N_k}$, where $N_k \geq 1$ |
| $W$ | The weight matrix of the model. | Matrix | $W \in \mathbb{R}^{k \times d}$ |
| $A_k$ | Local aggregator no | Integer | $A_k = D_k$ |
| $GS$ | Global server ID | Integer | $GS \geq 1$ |
| $C = \{C_1, C_2, \ldots, C_k\}$ | Pre-defined clients list object | Set of objects | $|C| = k$ |
| $ID_i$ | Client's ID no | Integer | $ID_i > 1$ |
| $IP_i$ | IP Address | String | Must be in a valid IPv4 or IPv6 format |
| $PC_i$ | Public certificate | String | Must conform to a standard format (*e.g.*, X.509) |
| $CP$ | Client program code | File object | Must be a script file |
| $HCP$ | Pre-defined FL client program hash | String | Valid cryptographic hash format (*e.g.*, SHA-128) |
| $HACP$ | Pre-defined client's attestation program hash | String | Valid cryptographic hash format (*e.g.*, SHA-128) |
| r | FL training round | Integer | $r \geq 1$ |
| $W_{r=0}$ | Initial model weight | Matrix | Dimensions must match model architecture, often initialized to zeros. |
| $W_r^k$ | Client's model weight at round | Matrix | Dimensions must match model architecture |
| $E$ | The total number of training epochs. | Integer | $E \geq 1$ |
| $\eta$ | Learning rate | Real | $0 < \eta < 1$ |
| $\beta$ | Batch size | Integer | $1 \leq \beta \leq N$ |
| $Pt$ | Backdoor pattern threshold | Integer | $1 \leq Pt < \mathbb{X}_k$ |
| $AP_{ij}$ | Client's attestation parameters | Set of objects | Contains attestation parameters (*e.g.*, $VC_k$, $HCP$) |
| $\tilde{f}_{Test}(.)$ | Test model | Object | TensorFlow/PyTorch saved model |
| $\tilde{f}_{Domain}(.)$ | Domain-level model | Object | TensorFlow/PyTorch saved model |
| $\tilde{f}_{Global}(.)$ | Global model | Object | TensorFlow/PyTorch saved model |
| $\tilde{f}_{Eval}(.)$ | Evaluation model | Object | TensorFlow/PyTorch saved model |
| $HPC_k$ | Client's public certificate hash | String | Valid cryptographic hash format (*e.g.*, SHA-128) |
| $HCD_k$ | Clients' local dataset hash | String | Valid cryptographic hash format (*e.g.*, SHA-128) |
| $VC_k$ | Client's verification list | Set of objects | Contains $PS$, $OP$ & $BS$ |
| $HVC_k$ | Verification list hash | String | Valid cryptographic hash format (*e.g.*, SHA-128) |
| $HCP_k$ | FL client program hash | String | Valid cryptographic hash format (*e.g.*, SHA-128) |
| $HACP_k$ | Client's attestation program hash | String | Valid cryptographic hash format (*e.g.*, SHA-128) |
| $PS$ | List of running processes | List of strings | – |

| Variable | Description | Type | Constraints/Range |
|---|---|---|---|
| $OP$ | List of open port | List of integers | Each element must be within $1 \leq Port \leq 65{,}535$ |
| $BS$ | Client's backdoor status | Boolean | True or False |
| $F1_k$ | Client's test model F1-score | Real | $0.0 \leq F1_k \leq 100.0$ |
| $T_k$ | Client's trust score | Integer | $T_k \geq 0$ |
| $\tau_{D_i}$ | List of client's trust score in domain | List of integers | – |
| $HAT_{Dk}$ | Local aggregator token for each domain | String | Must be valid token with SHA-128 format |
| $HT_{Dk}$ | Client node verification token | String | Must be valid token with SHA-128 format |
| $UC_k$ | Untrusted client list | List of $ID_i$ | Must be valid client ID |
| $UR$ | Dataset upload request | Boolean | True or False |
| $HDT_{UC_k}$ | Delegation token for untrusted client | String | Must be valid token with SHA-128 format |
| $Gt$ | GAN attack class threshold | Integer | $Gt \geq 1$ |
| $CW_k$ | Client's weightage list | List of integers | Typically, all elements sum to 1 $\sum CW_k = 1$ |
| $h$ | Suspected GAN client list | List of | Each element must be a valid client ID |
| $AGG_{Best}()$ | Selected best aggregation algorithm | Object | Aggregation algorithm's function (*e.g.*, FedAvg, MM, Krum, FedYogi) |
| **Datasets and preprocessing & attack scenarios** | | | |
| $x_i$ | The packet byte feature for the i-th sample | Vector | $x_i \in \mathbb{R}^d$ |
| $C_i$ | Traffic class | Integer | $1 \leq C_i \leq C$, where C is the total number of classes. |
| $y_i$ | The true label for the i-th sample | Integer | $y_i \in \{0, 1 \ldots C\}$ |
| $X_i$ | The packet byte vector | Vector | $x_i \in \mathbb{R}^d$ |
| $\mathbb{X}$ | Packet byte matrix containing feature and label (Dataset) | Matrix | $\{(x_i, y_i)\}_{i=1}^{N_k}$, where $N_k \geq 1$ |
| $\mathbb{X}_k$ | Dataset shard for local client | Set | $\mathbb{X}_k = \{(x_i, y_i)\}_{i=1}^{N_k}$, where $N_k \geq 1$ |
| $f$ | Function | Object | – |
| $\Delta W_m$ | Model weight | Matrix | $W \in \mathbb{R}^{k \times d}$ |
| $\alpha$ | Gradient negative factor | Real | $\alpha < 0.0$ |
| $Y_{Target}$ | Target class no (*e.g.*, 1 email, 2 FB Audio) | Integer | $Y_{Target} \geq 0$ |
| $M_{Malicious}$ | Malicious class no | Integer | $M_{Malicious} \geq 0$ |
| $m$ | Poison sample no | Integer | $m \geq 0$ |
| $X_B$ | Backdoor pattern | Vector | $0 \leq X_B \leq X_i$ |
| $c$ | Current features iteration | Integer | $0 \leq c \leq x_i$ |
| $Y_{Random}$ | Randomly selected of rows of label | Vector | Each element is a valid label $y_i$ |
| $P_{rate}$ | Poison sample rate | Real | $0.0 \leq P_{rate} \leq 100.0$ |
| $x_{Synth}$ | Synthetic packet byte | Matrix | Dimensions match feature space $x_{Synth} \in \mathbb{R}^{m \times d}$ |

server aggregates these poisoned updates, the global model's performance degrades. Model-canceling attacks aim to neutralize the global model during aggregation by setting the local model weights to zero (*Rey et al., 2022*). Another variant poisons the parameter updates with Gaussian noise (*Cao & Gong, 2022*). Meanwhile, a gradient

factor attack multiplies the local model gradient with a negative value to neutralize its training contribution (*Blanchard et al., 2017*). The MP attack typically has a broad and overt impact, as it tends to degrade the overall model accuracy significantly.

(c) ***Backdoor attack.*** This attack aims to cause the global model to misclassify specific traffic classes by implanting backdoor patterns during local model training (*Holodnak et al., 2022*). When the server aggregates the model parameters, it incorporates the backdoor pattern into the global model. During inference, the attacker can use a trigger pattern to make the model classify traffic to a specific outcome. For example, in an IDS setup, an attacker can use a backdoor to cause the IDS to classify malicious traffic as benign (*Nguyen et al., 2021*). Compared with LF and MP attacks, the backdoor attack is targeted and subtle because a successful attack should not noticeably degrade overall model accuracy.

(d) ***GAN-Based attack.*** This attack leverages a generative adversarial network (GAN) to generate synthetic traffic data, introducing classification bias into the global model. The attacker trains the GAN on traffic data similar to that of legitimate FL clients. The attacker then injects synthetic data with artificially incorrect labels into their training dataset (*Zhang et al., 2019*). This causes the model to become biased toward certain classes while maintaining overall accuracy. Furthermore, an attacker can also use GANs to generate a backdoor pattern and poison the FL model (*Zhang et al., 2021a*).

To mitigate and recover from the impact of adversarial attacks during FL training, various defensive measures have been proposed. The value-based approach examines the values of the parameters updated to filter extreme values or outliers during aggregation. For example, the median-mean (MM) method calculates the median for each parameter before averaging to exclude extreme values of the poison data. Another example is the trim-mean (TM) method, which removes a percentage of the lowest and highest values of the parameters before averaging to filter extreme values. In *Rey et al. (2022)*, both methods have been used to mitigate adversarial attacks during FL training. While the norm clipping method clips the gradient if it exceeds the threshold value set by the administrator (*Cao & Gong, 2022*). However, the effectiveness of the value-based approach diminishes with multiple numbers of attackers and may only work for specific types of attacks, as demonstrated in *Rey et al. (2022)*.

The distance-based approach calculates distances between updates to identify and exclude those that deviate significantly from the majority, which could indicate malicious intent. One example of a distance-based approach is Krum (*Blanchard et al., 2017*), where it works by selecting the most reliable update for aggregation by calculating pairwise Euclidean distances between updates. Multi-Krum extends it by selecting several of the least distant updates. However, Krum assumes that malicious participants constitute less than 33% of total clients, which makes it less effective with a large number of attackers. *Shi et al. (2023)* and *Fung, Yoon & Beschastnikh (2018)* calculates the Euclidean distance between models and considered scenarios such as benign *vs* benign and malicious *vs* benign. The detection of malicious updates occurred by analyzing the historical updates between the models.

In LFGurad (*Sameera et al., 2024*), the method employed Multi-Class SVM (MCSVM) for detecting malicious updates, a process that involved calculating the distance from each sample point to various class-specific hyperplanes. SRFL proposed by *Cao et al. (2024)* further enhances distance-based approaches by utilizing a trusted execution environment (TEE) and aggregation method, which clusters client updates based on the calculated distance between shared representations and membership degree. However, the distance-based approach is sensitive to parameters with high variance, which leads to higher false positives. Moreover, there is a lack of evidence that the approach can mitigate model poisoning and GAN-based attacks.

Researchers increasingly use hardware-based security approaches, such as trusted platform modules (TPM) and TEE, to secure the FL process. This trend occurs because most attacks involve tampering with either the datasets or the FL code. Thus, hardware-based approaches provide foundational trust for FL systems by protecting the integrity and confidentiality of FL components on individual client devices. Clients can use a TPM to securely store their unique cryptographic keys, sign parameter updates, or ensure they run untampered code.

For example, *Huang et al. (2022)* used a TPM for remote verification of data integrity and for storing results. However, a TPM only protects data-at-rest, while the system processes data in the main host memory or CPU. In contrast, a TEE offers more comprehensive protection by creating an isolated enclave where the system can execute the FL process. Following this approach, *Cao et al. (2024)* leverage a TEE to safeguard sensitive FL components from tampering, and *Muhr & Zhang (2022)* use TEE to shield local client updates. However, TEE requires a specialized CPU with features that are typically only found in enterprise-grade hardware. A TPM, on the other hand, is more readily available, and operating systems like Windows 11 now make it a default installation requirement.

Meanwhile, the weighted averaging (WA) approach aims to limit the influence of malicious updates by assigning different weights to clients and often use in tandem with other approaches such as distance-based detect malicious updates. In the pFL-IDS (*Thein, Shiraishi & Morii, 2024*), clients were reweighed based on a normalized similarity score, where the malicious client had a larger deviation from the global model value. However, the effectiveness of the WA scheme relies on the server's ability to identify malicious clients and assign the correct weight values accurately. There are several methods to identify malicious clients, such as calculating model distances of the neighbour (*Shi et al., 2023*) or outlier detection (*Xu et al., 2022*). However, its effectiveness is limited to specific types of attacks and datasets, as shown in *Zhou et al. (2022)*. Moreover, the approach only reduces the effectiveness of a backdoor attack for specific datasets instead of preventing it altogether, especially if it involves multiple attackers (*Xie et al., 2020*).

*Raza et al. (2022)* proposed another defensive approach that uses anomaly detection, creating a reference model trained on a public dataset. If the discrepancy between the reference and auditor models exceeds a threshold, the framework flags the update as malicious and removes it from aggregation. However, this framework cannot detect small, colluding poisoning attacks or more advanced threats like backdoor or GAN-based attacks. Furthermore, this detection method may fail when a relevant public dataset is

unavailable. In another work, *Guo, Xu & Zhu (2023)* introduced a sign-based approach to detect poisoning attacks in parameter updates. The sign of a gradient indicates its direction of change (*i.e.*, whether it is positive or negative). However, the sign-based approach might fail to detect subtle yet harmful manipulations that do not alter an update's direction. Moreover, the authors did not evaluate its effectiveness against GAN-based attacks.

Table 3 summarizes related works by mapping each work's approaches, methods, applications, datasets, attack evaluations, security features, strengths, and weaknesses. The table also lists the adversarial attacks that the original authors used for evaluation and demonstrates the reported effectiveness of each approach. In summary, most existing defenses effectively counter only specific, rather than multiple, types of attacks. The effectiveness of these approaches also diminishes as the number of attackers increases, especially when malicious clients collude. Additionally, current approaches have several other limitations, including a high risk of false positives, a dependency on public datasets, and the potential for sign-based methods to miss subtle attacks. Furthermore, researchers did not explicitly design most proposed defenses for the FL-based NTC workflow; consequently, these defenses lack features such as hardware-based security, communication privacy safeguards, and client identity verification. Therefore, future work should focus on enhancing defensive effectiveness against multiple adversarial attack types in FL-based NTC.

## FL-BASED NTC ARCHITECTURE

This section describes the architecture and training process for our FL-based NTC model. Using FL for training a deep learning (DL) based NTC model shifts the paradigm from centralized to distributed training. Figure 1 shows a typical FL-based NTC training architecture, which this study adapts from the work of *Ariffin, Zaki & Anuar (2023)*, *Guo & Wang (2023)*, *Popoola et al. (2022)* and *Wang et al. (2018)*.

FL-based NTC training uses a centralized topology that connects multiple edge devices (FL clients) to a central aggregation server. In this architecture, all FL clients send their parameter updates directly to the server. Each edge device, such as a firewall or gateway router, operates within an organization's boundary to provide network services. Because these devices forward traffic, they can collect and preprocess data from local endpoints like workstations, laptops, and Internet of Things (IoT) devices.

The edge devices use this local data and various DL algorithms to train their own NTC models. After local training, each device uploads its model parameters to the central server. The server then aggregates the parameters from all clients to build and refine a global model. Finally, the server distributes the updated global model parameters back to the edge devices. This collaborative approach allows different organizations to build a robust, shared model without exchanging sensitive local data. The entire FL-based NTC training process, which Fig. 1 depicts, consists of the following steps:

(1) ***Traffic data collection and preprocessing (Client).*** Before training begins, the FL clients need to collect traffic data by capturing the raw packet bytes from the network interface responsible for forwarding traffic to the local endpoint devices. After that, the

**Table 3  Summary of related works.**

| References | Defensive approaches | Methods | Application areas | Datasets | Adversarial attacks evaluation | | | | Security features | | | Strengths | Weaknesses |
|---|---|---|---|---|---|---|---|---|---|---|---|---|---|
| | | | | | LF | MP | Backdoor | GAN | Hardware-based security | Encrypted communication | Identity verification | | |
| *Blanchard et al. (2017)* | Distance-based | Krum Multi-Krum | Image classification | MNIST CIFAR | ✗ | ✓ | ✗ | ✗ | ✗ | ✗ | ✗ | Work with the black box model without prior trust assumption. | Only focus on model poisoning attacks. It works on the assumption that malicious participants make up less than 33% of the total. |
| *Xie et al. (2020)* | Distance-based, weighted-averaging | Krum Fools golds | Image classification | LOAN MNIST CIFAR Tiny-ImageNet | ✗ | ✗ | ✓ | ✗ | ✗ | ✗ | ✗ | Introduce a more effective distributed backdoor attack. Evaluate the backdoor attack effectiveness against multiple defensive approaches. | The focus is only on backdoor attacks, and little is discussed about how to mitigate distributed backdoor attacks. |
| *Muhr & Zhang (2022)* | Hardware-based security | TEE Homomorphic encryption | Image classification | Fashion-MNIST CIFAR | ✓ | ✓ | ✗ | ✗ | ✓ | ✓ | ✓ | Provide privacy preservation and identity verification. | Only designed to mitigate LF and MP attacks |
| *Raza et al. (2022)* | Anomaly detection | References model, one-class support vector machine | Healthcare | MIT-BIT arrhythmia | ✓ | ✓ | ✗ | ✗ | ✗ | ✗ | ✓ | Effective against multiple variants of MP attacks. | Effectiveness is dependent on the availability of public datasets. Unable to detect small colluding attacks. Lack of evaluation against backdoor and GAN-based attacks. |
| *Rey et al. (2022)* | Value-based | Median-mean Trim-mean Sampling | Network traffic classification, IDS | N-BaIoT | ✓ | ✓ | ✗ | ✗ | ✗ | ✗ | ✗ | Highlight the importance of scalable defence against multiple attackers. | Lack of evaluation against backdoor and GAN-based attacks. |
| *Shi et al. (2023)* | Distance-based, weighted-averaging | Analyse statistical distance to select benign clients. | Image classification | MNIST | ✓ | ✗ | ✗ | ✗ | ✗ | ✗ | ✗ | Evaluate against trim-mean and sniper defensive measures. Effective against a variant of LF attacks. | Lack of evaluation against MP, backdoor, and GAN-based attacks. Effectiveness depends on selecting the right benign client, and malicious clients can overwhelm the training. |
| *Zhou et al. (2022)* | Weighted-averaging variant | Weight-based anomaly detection, differential privacy | Image classification | MNIST fashion-MNIST CIFAR-10 | ✗ | ✓ | ✗ | ✗ | ✗ | ✓ | ✗ | Include a differential privacy method to enhance the privacy of updated parameters. | Lack of evaluation against LF, backdoor, and GAN-based attacks. Effectiveness depends on the ability to assign weight to the clients appropriately. |

(Continued)

| References | Defensive approaches | Methods | Application areas | Datasets | Adversarial attacks evaluation | | | | Security features | | | Strengths | Weaknesses |
|---|---|---|---|---|---|---|---|---|---|---|---|---|---|
| | | | | | LF | MP | Backdoor | GAN | Hardware-based security | Encrypted communication | Identity verification | | |
| Xu et al. (2022) | Anomaly detection, filtering | Clustering to filter outlier | Image classification | IRIS wine quality | ✓ | ✓ | ✗ | ✗ | ✗ | ✗ | ✗ | Effective against multiple poisoning attacks | Lack of evaluation against backdoor and GAN-based attacks. Higher computational requirement due to clustering. |
| Cao & Gong (2022) | Value-based | Norm clipping | Image classification text classification | MNIST fashion-MNIST purchase | ✗ | ✓ | ✗ | ✗ | ✗ | ✗ | ✗ | It uses fake clients to increase attack effectiveness. Evaluate against median, trimmed mean, and norm clipping method. | Effectiveness decreases as the number of attackers increases. Lack of evaluation against LF, Backdoor, and GAN-based attacks. |
| Guo, Xu & Zhu (2023) | Sign-based | FedSIGN: poisoning detector | Image classification | MNIST F-MNIST CIFAR10 | ✓ | ✓ | ✓ | ✗ | ✗ | ✗ | ✗ | Evaluated LF, MP, and backdoor attacks involving multiple adversaries. | Lack of evaluation against GAN-based attack. Reduce effectiveness in detecting subtle attacks. |
| Cao et al. (2024) | Distance-based hybrid | Trusted execution environment SRFL client clustering *via* MMRA-MD | Image classification | MnistNet Cifar10Net Cifar100Net | ✗ | ✗ | ✓ | ✗ | ✓ | ✓ | ✗ | Utilize hardware-based security to mitigate tampering. | Lack of evaluation against LF, MP, and GAN-based attacks. Sensitive to high variance parameters update, leading to higher false positives. Soft clustering leads to low stability and fluctuation during convergence. |
| Sameera et al. (2024) | Distance-based | LFGurad multi-class SVM | Internet-of-vehicles | Fashion MNIST | ✓ | ✗ | ✗ | ✗ | ✗ | ✗ | ✗ | Doesn't require client compliance Effective against multiple attackers | Only designed to mitigate LF attacks. |
| Thein, Shiraishi & Morii (2024) | Weighted-averaging variant | pFL-IDS normalized weight | Network traffic classification, IDS | N-BaIoT | ✓ | ✓ | ✗ | ✗ | ✗ | ✗ | ✗ | Effective against LF and MP attacks with multiple attackers. | Only designed to mitigate LF and MP attacks Only effective if able to identify the malicious client correctly. |
| This study SHeRAA-FL | Hybrid | Remote attestation scoring, hierarchical training, adaptive aggregation | Network traffic classification | ISCX-VPN 2016 | ✓ | ✓ | ✓ | ✓ | ✓ | ✓ | ✓ | Provide consistent protection against multiple types of adversarial attacks, even with multiple attackers. | Based on the Flower framework. They were only evaluated on NTC workflow. |

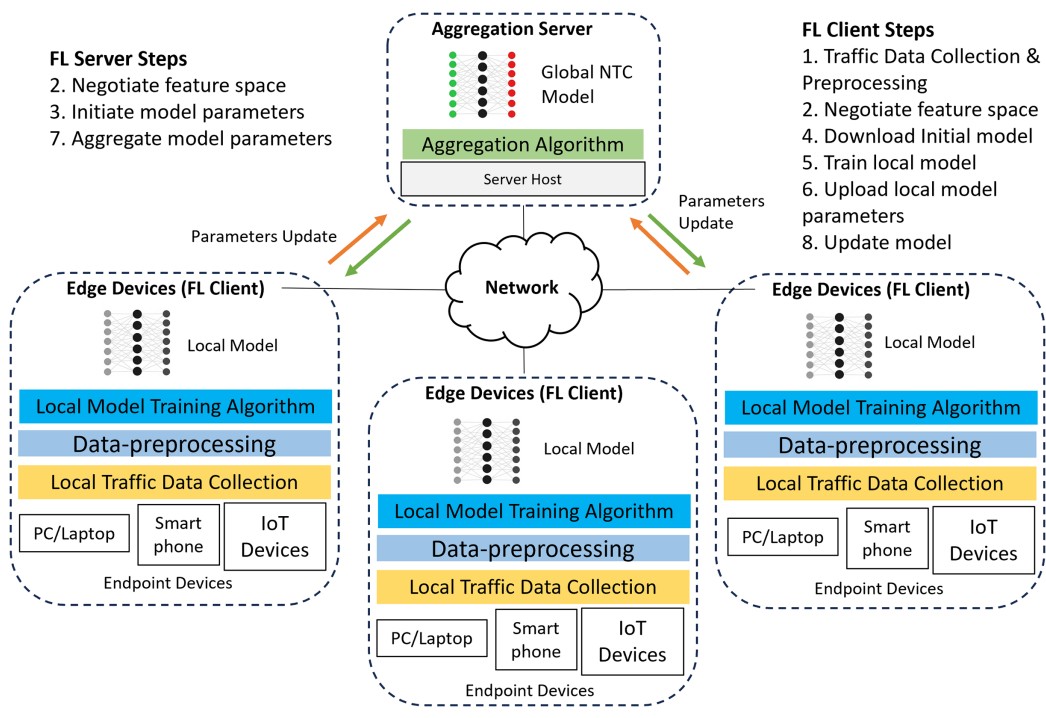

**Figure 1** **Typical FL-based NTC training architecture.**

raw packet bytes need to be saved into packet capture (PCAP) files for easier processing. These raw packet bytes undergo preprocessing before being used in local model training. The preprocessing method used by this study is discussed in the methodology section.

(2) ***Negotiate feature space (Server-Client).*** The server and clients negotiate a common feature space, $X = (x_1, x_2, \ldots, x_n)$, using a horizontal FL setup where different edge nodes share the same feature space but possess distinct traffic samples. Each feature, $x_i$, represents the value of the $i$-th packet bytes. Therefore, agreeing on the same feature space is crucial before collaborative training begins between the server and FL clients.

(3) ***Initiate model parameters (Server).*** The server initiates the process by creating an initial model based on various types of DL algorithms. This study uses a multilayer perceptron (MLP) and, in the initialization steps, sets the model with an empty weight vector $W = 0$. It then sets the hyperparameters required for FL training, including the number of rounds $r$, epochs $E$, learning rate $\eta$, and batch size $\beta$.

(4) ***Download initial model parameters (Client).*** The edge clients download the initial model along with the hyperparameters and feature space set by the server.

(5) ***Train local model (Client).*** Using the downloaded initial model $W_r^k = 0$ and hyperparameters, each edge client trains the local model with its local dataset $\mathbb{X}_k$. The local model training is conducted based on DL algorithm chosen by the server when initiating the model parameters. During this local training step, a malicious client can manipulate its training data or model weights to poison the local model. The server

cannot control this process because each client performs its training locally and independently.

(6) ***Upload local model parameters (Client).*** The edge clients upload the weights $W_{r+1}^k$, of their trained local models to the server for aggregation with parameters from other local models. During this step, a malicious client will send a poisonous weight update to the server, intending to corrupt the aggregation process and the resulting global model.

(7) ***Aggregate model parameters (Server).*** The server aggregates the local model weights using the Federated Averaging algorithm $W_{avg} = \sum_{i=1}^{k} \frac{n_i}{N} W_i$ where $W_1, W_2, \ldots, W_k$ represent the weights of the models from edge clients $k$, each with the local dataset $\mathbb{X}$ size $n_k$ and $N$ is the total dataset size $N = n_1, n_2, \ldots, n_k$. The server then distributes the aggregated model weight back to all clients as $W_{r+1}^k \rightarrow$ all k. The typical federated averaging (FedAvg) algorithm provides good aggregation performance, however it doesn't provide protection against adversarial attacks. Thus, robust aggregation algorithm such trim-mean or Krum are needed to mitigate the impact of adversarial attacks during model aggregation.

(8) ***Update local model (Client).*** Each client updates its local model with the new aggregated weights from the server. The clients repeat this training and updating cycle (steps 4–8) for a predetermined number of communication rounds $r$. This iterative process allows each client to indirectly learn from the traffic patterns of other organizations, which enhances the model's overall accuracy and generalization. Once the process completes the maximum number of rounds, the FL training concludes. The server then distributes the final global NTC model to all edge clients for inference. However, without effective defensive measures, the server will inadvertently distribute a poisoned global model to all participating clients, compromising the entire network.

## METHODOLOGY

This section details the design of our proposed framework, which secures the FL-based NTC training process against various adversarial attacks. The framework uses three core mechanisms to detect and mitigate these threats: remote attestation scoring, hierarchical training and adaptive aggregation mechanism.

Additionally, this section discusses the methodology for evaluating the effectiveness of the proposed SHeRAA-FL framework in mitigating multiple types of adversarial attacks. The discussion includes the federated learning testbed setup, network traffic datasets, data preprocessing, and attack scenarios. Figure 2 shows this study's research flowchart. The research starts with a literature review to identify the problems and limitations of existing adversarial defenses for FL-based NTC. Based on these limitations, this study designs a framework to provide enhanced defense against multiple types of adversarial attacks. After designing the framework, we implemented it using Python and related libraries on a TPM-enabled host.

To evaluate the framework's effectiveness, this study set up a federated learning testbed for NTC training, which we explain in the following subsection. We must pre-process the

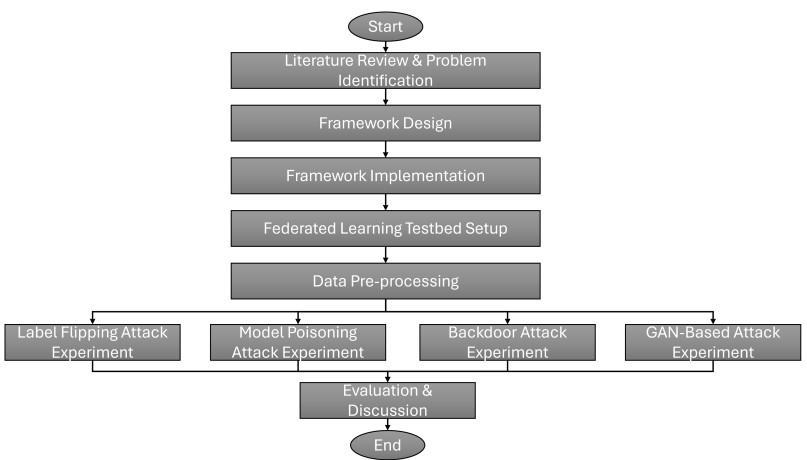

**Figure 2** **Research flowchart of this study.**

raw packets in the datasets before we can begin the evaluation. We explain these pre-processing steps in the next subsection. For the evaluation, this study conducted four main experiments, one for each adversarial attack type: label flipping, model poisoning, backdoor, and GAN-based attacks. Each experiment uses a similar setup with five scenarios: a normal (control) scenario, and scenarios with one, two, three, and four attackers.

In each of the attack scenarios, this study used several defensive aggregation algorithms as benchmarks. These algorithms represent different defensive approaches against adversarial attacks in FL. The algorithms include:

(a) **FedAvg.** This is the standard "vanilla" FL aggregation approach.

(b) **Weighted averaging (WA).** This approach limits the influence of malicious updates. In this study experiment, we assumed the server had already identified the malicious clients and assigned a lower weight (20%) to their parameter updates. Previous studies by *Zhou et al. (2022)* and *Shi et al. (2023)* used similar variants to mitigate adversarial attacks.

(c) **Median (MM) and trimmed mean (TM).** These value-based approaches defend the FL training by excluding extreme (outlier) values from the client updates during aggregation.

(d) **Krum and Multi-Krum.** These distance-based approaches filter outlier updates by calculating the distance between various client updates and excluding those that are too far from the main cluster.

Many studies use MM, TM, Krum, and Multi-Krum as standard benchmarks for evaluating FL defenses (*Cao et al., 2024*; *Fung, Yoon & Beschastnikh, 2018*; *Sameera et al., 2024*; *Rey et al., 2022*; *Thein, Shiraishi & Morii, 2024*). This study also benchmarked the framework with other approaches such as LFGurad (*Sameera et al., 2024*), FoolsGold (*Fung, Yoon & Beschastnikh, 2018*), pFL-IDS (*Thein, Shiraishi & Morii, 2024*) and SRFL

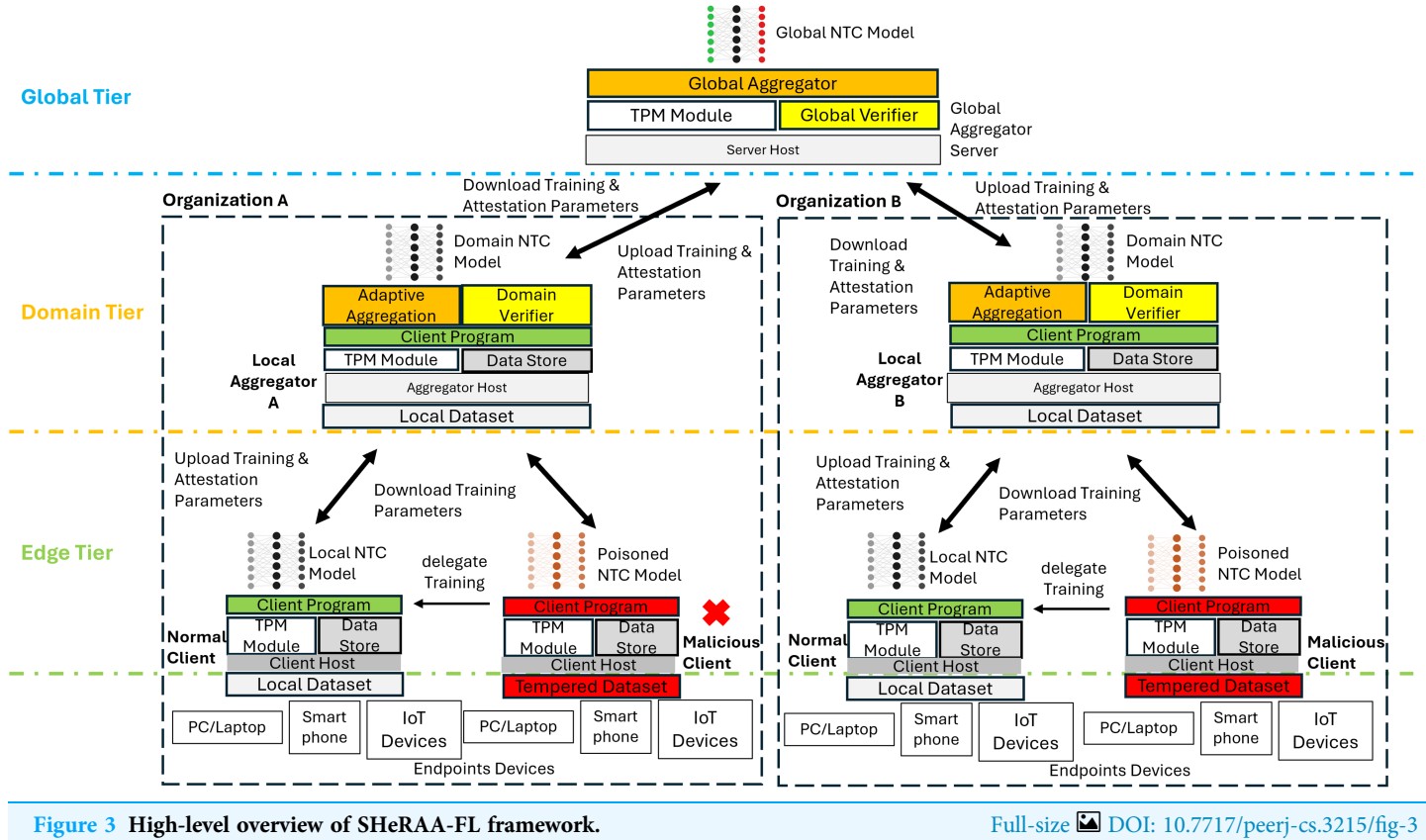

**Figure 3** High-level overview of SHeRAA-FL framework.

(*Cao et al., 2024*). After collecting the experimental results, we evaluated and discussed the framework's effectiveness. The evaluation also discusses the computational overhead of the framework.

## Secure framework design

This study proposes the SHeRAA-FL framework to enhance the defensive measures against multiple adversarial attacks for FL-based NTC. The framework consists of three main mechanisms: remote attestation scoring, hierarchical training, and adaptive aggregation. Figure 3 shows a diagram of a high-level overview of SHeRAA-FL.

The proposed framework clusters the FL clients based on domain in a hierarchical topology to enhance data privacy. Each edge client belongs to a domain, and multiple domains $D_k$ can join the FL training. Each client within the domain belongs to the same organizational or administrative boundary. For example, a university campus network has two edge devices participating in FL, where each edge device serves as a gateway router for different faculty. Thus, both edge devices are clustered in the same domain. Besides that, the framework organizes the hosts in a hierarchical topology, which has global, domain, and edge tiers. The edge is the lowest tier, which contains FL edge clients such as firewalls, switches, or routers that filter and forward endpoint traffic. The client captures a sample of the endpoint packets and uses it as a local dataset $\mathbb{X}_k$ for training. The clients have a TPM and data store module for storing attestation-related data. They also have a program

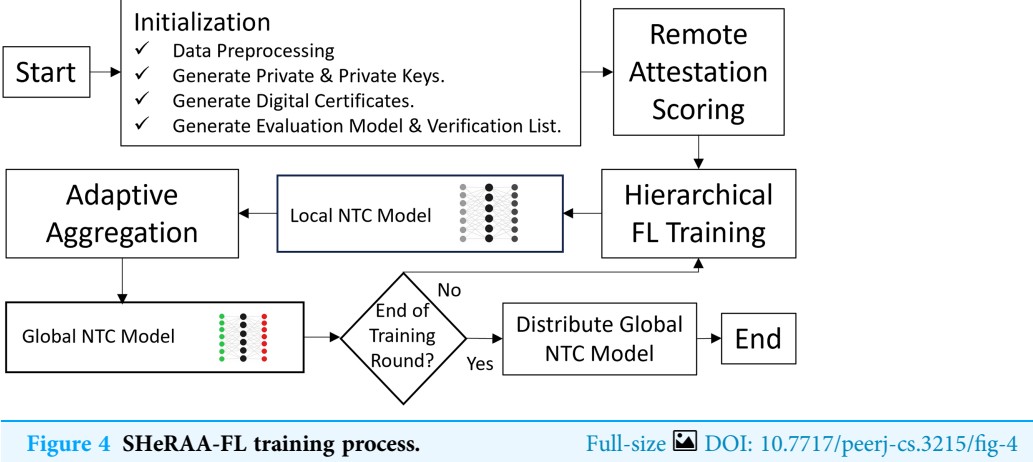

**Figure 4  SHeRAA-FL training process.**

containing instructions on training local models and participating in the attestation and FL process.

Meanwhile, the domain tier contains a local aggregator $A_k$ which verifies and aggregates the parameters $\{AP_{kj}, W_r^k\}$ of FL clients within the domain $D_k$. The local aggregator has all the modules of normal clients with the addition of a domain verifier and an adaptive aggregation module. Besides that, local aggregators also perform the training tasks of normal clients. The local aggregator produces a domain-level model $\tilde{f}_{Domain}(.)$ by aggregating the weight parameters of clients within its domain. Lastly, the global tier contains a global aggregator server with TPM, a global verifier, and an aggregator module. The global aggregator produces a global model $\tilde{f}_{Global}(.)$ by aggregating the parameters of the domain model sent by the local aggregator of each domain.

Figure 4 outlines the flow of the FL training process within the SHeRAA-FL framework. The process begins with an initialization stage where the clients perform data preprocessing on their local datasets. Next, the global server and the clients generate public and private keys to create digital certificates. The framework requires these certificates to establish encrypted TLS communication and verify host identities. After that, the clients train an evaluation model and create a verification list, both of which are necessary for the remote attestation process. The server and clients then engage in a remote attestation scoring process. Then, the framework starts with the hierarchical training and adaptive aggregation process, which continues until the process reaches the maximum training round. Once training is complete, the server distributes the global NTC model to the clients for inference.

The following are details on remote attestation scoring, hierarchical training, and adaptive aggregation mechanisms.

(a) *Remote attestation scoring.* Before FL training began, the framework's first step was establishing trust among the FL server and clients *via* the remote attestation scoring process. Algorithm 1 shows the framework's remote attestation scoring process. The proposed framework leverages hardware-level security, such as TPM, to store server

---

**Algorithm 1** Remote attestation scoring process.

**Require:** Pre-defined clients list $C = \{C_1, C_2, \ldots, C_k\}$ containing clients' identification number $ID_i$, domain ID $D_i$, IP address $IP_i$, and public certificate $PC_i$ for the client $i = 1, 2, \ldots, k$. Pre-defined FL client program hash $HCP$, client attestation program hash $HACP$, initial model weight $W_{r=0}$, hyperparameters {training round r, epochs $E$, learning rate $\eta$, batch size $\beta$}, Backdoor pattern threshold $Pt = 5,000$;

1. **Client Initialization:** Each client $k$ prepare attestation parameters $AP_{ij}$ which contains a test model $\tilde{f}_{Test}(.)$,

2. FL program hash $HCP_k$, attestation program hash $HACP_k$, public certificate hash $HPC_k$,

3. local dataset hash $HCD_k$, verification list $VC_k$ and verification list hash $HVC_k$.

4. Each client $k$ train $\tilde{f}_{Test}(.)$ using $\mathbb{X}_k$, and calculating the hashes $HCP_k$, $HACP_k$, $HPC_k$, and $HCD_k$.

5. Each client $k$ Generate $VC_k = (PS, OP, BS)$ verification list and check potential backdoor in local dataset.

6. $PS$ containing number of running processes, $OP$ number of open ports and $BS$ client's backdoor status.

7. **Client Attestation Program Check for Backdoor Pattern:**

8. **For** each row in $\mathbb{X}_k$ do:

9. **If** bytes in row not in recurring_pattern: recurring_pattern.append(row);

10. **Else:** PatternCount++;

11. **End For**

12. **If** PatternCount >= $Pt$: backdoorStatus $BS$ = **True** and remove pattern;

13. After client $k$ finished with $VC_k$, then generate $HVC_k$.

14. all clients $k$ establish TLS connection to the global server using server $PC_{GS}$ and upload all $AP_{ij}$

15. **Global Server GS Verify Uploaded Parameters:**

16. **If** client $k$ $\{ID_k \mid D_k \mid IP_i \mid HACP_k \mid HPC_k\} \neq C_k$ stored by the server:

17. Drops client $AP_k$ and remove client $k$ from FL network;

18. **Else** store $AP_{kj}$ in TPM:

19. $\{HCP_{ID_i}, HACP_{ID_i}, HCD_{ID_i}, HVC_{ID_i}, HPC_{ID_i}\} \rightarrow TPM(GS)$ for $ID_i = 1, 2, \ldots, k$

20. **Attestation Scoring Process:**

21. **For** each domain $D_k$:

22. finds $PS_{min} = \min(PS_1, PS_2, \ldots, PS_k)$; $OP_{min} = \min(OP_1, OP_2, \ldots, OP_k)$ from

23. all $VC_k$ in domain $D_k$, then evaluate $\tilde{f}_{Test}(.)_k$ using $\mathbb{X}_T$ test dataset and finds

24. $F1_{max} = \max(F1_1, F1_2, .., F1_k)$ the model with the highest accuracy;

25. **For** each client $k$ in domain $D_k$ :

26. **If** client $HCP_k = HCP$: add client $k$ score $T_k = T_k + 15$;

27. **Else:** $T_k = T_k + 0$; Client $k$ become $UC_k$ untrusted client.

28. **If** $PS_k = PS_{min}$ **Or** $OP_k = OP_{min}$: add client $k$ score $T_k = T_k + 5$;

29. **Else:** $T_k = T_k + 0$.

30. **If** client $BS$ = **False**: add client $k$ score $T_k = T_k + 5$;

31. **Else:** $T_k = T_k + 0$.

32. **If** $F1_k = F1_{max}$: add client $k$ score $T_k = T_k + 10$;

33. **Else If** $F1_k >= 60\%$: add client $k$ score $T_k = T_k + 5$;

34. **Else:** $T_k = T_k + 0$;

35. **End For**

---

| Algorithm 1 (continued) |
| --- |

36. **End For**

37. **Local Aggregator Selection Process:**

38. **For** each domain $D_k$:

39. Finds highest trust score $T_k$ from all $\tau_{D_i}$ in domain $D_k$; $T_{max} = \max(T_k)$;

40.     **For** each client $k$ in domain $D_k$:

41.         **If** $T_k = T_{max}$: select $k \rightarrow A_k$ as an aggregator for domain $D_k$; Server generates

42.            $HAT_{Dk}$ aggregator token, store in TPM, and send the token to client $k$;

43.            The server also send attestation parameters $\{HAT_{Dk}, \tau_{D_i}, HCP, HACP, HT_{Dk}, AP_{Dk}, UC_{Dk}\} \rightarrow A_k$

44.            along with $W_{r=0}$, hyperparameters, trust score and untrusted client list. Then $A_k$ store in TPM.

45.         **Else:** The Global Server generates $HT_{Dk}$ node verification token, store in TPM and

46.            **If** client $k$ in untrusted client $UC_k$ list:

47.                send the token to client $k$ along with dataset upload request $UR$ and $\{HT_{Dk}, IP_{A_D}, PC_{A_D}\} \rightarrow k$;

48.            **Else:**

49.                send the token to client $k$ along with $\{HT_{Dk}, IP_{A_D}, PC_{A_D}\} \rightarrow k$;

50.            client $k$ store the parameters in TPM $\{HT_{Dk}, HCD_k\} \rightarrow TPM(k)$;

51.     **End For**

52. **End For**

and client FL program and verification parameters in a secure enclave to minimize the risk of tampering during the attestation and training process. This study selected the TPM for this study due to its wide availability in commodity computing hardware. In contrast, a TEE requires specialized processors that include features like Intel SGX or ARM TrustZone.

The process starts with the initialization stage, where each client $k$ generate and prepares attestation parameters $AP_{ij}$ which are needed by the global server $GS$ to calculate client $k$ trust score $T_k$. The attestation parameters $AP_{ij}$ contains test model $\tilde{f}_{Test}(.)$ which are trained using client $k$ local dataset $\mathbb{X}_k$, local dataset hash $HCD_k$, verification list $VC_k$, a hash of the client's FL program $HCP_k$, attestation program $HACP_k$, public certificate $HPC_k$, and verification list $HVC_k$. The training of the test model only involves several epochs to shorten the training time, as it is only used by $GS$ to evaluate the dataset quality. Meanwhile, for the $VC_k$, the clients' attestation program provides the number of running processes $PS$ and open port $OP$ on the host. Besides that, the clients' attestation program also checks $\mathbb{X}_k$ from potential backdoor patterns. If the recurring pattern exceeds the threshold $Pt$ value, the attestation program sets the backdoor status $BS$ as True, remove the identified pattern and include the information in $VC_k$. The threshold value is hardcoded in the remote attestation program, and tampering with it causes the program's hash value to differ from the global server reference value.

After initialization, clients $k$ send the $AP_{ij}$ to $GS$ *via* TLS to ensure communication is kept private and the client communicates with the verifiable global server to avoid

spoofing. When *GS* receives the client $k$ $AP_{ij}$, first, it verifies the information by comparing it with the pre-defined client list $C = \{C_1, C_2, \ldots, C_k\}$. If there is a discrepancy with the client's $ID_k$, domain number $D_k$, IP Address $IP_k$, public certificate $HPC_k$, or the attestation program hash $HACP_k$ the server removes the suspected client data from training. Tampering with the $AP_{ij}$ indicates an attempt to bypass the attestation process or initiate a Sybil attack, which involves creating multiple fake identities (*Xiao et al., 2022*).

Once verified, the valid clients' hash values are stored in the server's TPM to avoid tampering during the ongoing attestation process. The attestation scoring process is conducted per domain based on information in the attestation parameters $AP_{ij}$. First, client $k$ with verified FL client programs $HCP_k$ are given fifteen trust score $T_k$, while clients with tampered FL client programs are given zero points and added to untrusted client $UC_k$ list. Clients with tampered programs have a higher risk of being malicious, and attackers can manipulate the structure of the model and update parameters by tampering with the FL program. The proposed algorithm will take further action on clients with tampered datasets without resorting to simply discarding the potentially clean dataset. Second, clients with minimum *PS* or *OP* are given five points $T_k$ each, as clients with lower values have a lower risk of being compromised. Third, clients with no suspected backdoor pattern are given five trust points $T_k$.

The fourth criterion is the clients' test model $\tilde{f}_{Test}(.)$ performance. The client with the highest F1-score is given 10 points, while clients with an F1-score of more than 60% are given five points $T_k$. Higher test model performance is a good indicator that the client datasets have a lower risk of being poisoned, while models with more than 60% F1-score indicate the client has a reliable dataset for training the NTC model (*Jenefa & Edward Naveen, 2023*). The trust score assigned to each evaluation criterion is based on the severity of potential security risks. For example, tampering with the FL program has the highest severity; thus, a potentially malicious client is penalized fifteen points as tampering with the FL program enables attackers to modify the model structure, update parameters, and bypass security protocols. Meanwhile, clients with low test model accuracy are penalized with 10 to 5 points as it is a good indicator that the dataset has been poisoned. Moreover, the trust score value is also derived by experimentation during the design stage to ensure less risky clients are chosen as local aggregator $A_k$ and risky clients' trust score $T_k$ are penalized to reduce their influence during the adaptive aggregation mechanism.

After *GS* calculates the trust score $T_k$ for each client $k$ in the domain, the client with the highest $T_k$ in the domain is selected as the local aggregator $A_k$ for the domain $D_k$. To uniquely identify the $A_k$, *GS* generates an aggregator token $HAT_{Dk}$ and stores the token in TPM. The *GS* sends the token to $A_k$ along with other information, such as trust score $T_k$, FL Client program hash $HCP$, local dataset hash $HCD_k$, $AP_{Dk}$, verification token $HAT_{Dk}$ and untrusted client list $UC_{Dk}$ of all clients $k$ in its domain $D_k$, which are needed during the hierarchical training process. Meanwhile, for other clients $k$, if the client is untrusted $UC_k$, *GS* send verification token $HT_{Dk}$, local aggregator $IP_{A_D}$,

local aggregator public certificates $PC_{A_D}$, along with a dataset upload request $UR$ which will be used for the training delegation process during hierarchical training. For other client $k$ not in untrusted client $UC_k$, $GS$ only send a verification token $HT_{Dk}$, local aggregator $IP_{A_D}$, and local aggregator public certificates $PC_{A_D}$. Both local aggregators $A_k$ and clients store the information they receive in TPM to prevent tampering.

(b) *Hierarchical training.* After selecting local aggregator $A_k$ and assigning trust score $T_k$, the framework begins with the hierarchical FL training. Algorithm 2 shows the hierarchical training process of the framework. Client $k$, which received aggregator token $HAT_{Dk}$ assume the role of $A_k$ and listens for client requests *via* TLS. Meanwhile, others become local clients and establish communication with local aggregators *via* TLS using local aggregator IP address $IP_{A_D}$ and public certificate $PC_{A_D}$ received from global server $GS$ earlier. If clients $k$ received dataset upload request $UR$ from $GS$ during remote attestation process, the client requires them to upload their local dataset $\mathbb{X}_k$ to the selected $A_k$ of domain $D_k$ using its $ID_k$ and $HT_{ID_k}$ as identification.

Meanwhile, clients $k$ which doesn't receive $UR$ are only require to send the hash of their local dataset $HCD_k$ and their FL client program $CP$ to the selected $A_k$ of domain $D_k$ using its $ID_k$ and $HT_{ID_k}$ as identification. At this stage, only the suspected untrusted clients $UC_k$ are given a choice either to upload their local dataset $\mathbb{X}_k$ to the selected $A_k$ of domain $D_k$ or being removed from the FL training. To ensure data privacy, the local dataset $\mathbb{X}_k$ is uploaded only to $A_k$ in $D_k$, thus the local dataset $\mathbb{X}_k$ is not exposed to the host such as $GS$, $A_k$ or client $k$ outside the organizational boundary. Moreover, to avoid unnecessary usage of bandwidth, only untrusted clients $UC_k$ requires uploading local dataset $\mathbb{X}_k$ to $A_k$ which nearest to $UC_k$. This is to avoid simply discarding a good dataset although the client's FL program has been compromised.

After that, the selected $A_k$ perform verification process for clients $k$ in its domain $D_k$. The verification process is crucial to ensure the clients $k$ do not alter any data after the remote attestation process with $GS$. The initial steps involve the $A_k$ verifying the FL program $CP$ and local dataset hash $HCD_k$. The verification is done differently for untrusted and normal clients. If the client $k$ in untrusted client $UC_k$ list, first $A_k$ check the verification token $HT_{ID_k}$ and the uploaded $\mathbb{X}_k$. If the token mismatch from information received from $GS$ or clients $k$ refuse to upload the $\mathbb{X}_k$, $A_k$ removes the client from FL training along with its data. On the other hand, if the client $k$ upload $\mathbb{X}_k$ to $A_k$ secure storage, $A_k$ delegates the client's training task to other trusted clients.

The delegation process involves generating a delegation token $HDT_{UC_k}$ and new $ID_{UC_k}$, then sending the untrusted client's dataset $\mathbb{X}_{UC_k}$ to the selected trusted client. The new $ID_{UC_k}$ instance will inherit the trust score $T_k$ of the untrusted client $UC_k$. If there is no suitable, trusted client to delegate, the training task will be delegated to the local aggregator $A_k$. This will ensure the $UC_k$ local dataset $\mathbb{X}_k$ is not simply discarded and trained using a verified FL training program. Although there is a risk to the dataset $\mathbb{X}_k$ is poisoned by the malicious client, the risk will be minimized by an adaptive aggregation mechanism. Meanwhile for normal clients $k$, $A_k$ check the verification token $HT_{ID_k}$, local dataset hash $HCD_k$ and FL client program hash $HCP$. If any of the

---

**Algorithm 2 Hierarchical training process.**

**Require:** Aggregator $HAT_{Dk}$ token, node verification token $HT_{Dk}$, client trust score $\tau_{D_i}$, client attestation parameters $AP_{ij}$, local aggregator IP $IP_{A_D}$, public certificate $PC_{A_D}$, $W_{r=0}$, hyperparameters, untrusted client $UC_k$ list;

1. **If** client $k$ received $HAT_{Dk}$: $k \rightarrow A_k$ become a local aggregator in the domain $D_k$ and start running

2.     aggregator and client program, listen to client requests *via* TLS

3.     using $IP_{A_D}$ and $PC_{A_D}$;

4. **Else If** client $k$ received $HT_{Dk}$: Become a local client in the domain $D_k$ and start

6.     client program and use $IP_{A_D}$ *and* $PC_{A_D}$ to initiate a TLS connection with $A_k$ domain $D_k$,

7.     **If** client $k$ received dataset upload request $UR$:

8.       upload local dataset $\mathbb{X}_k$ to local aggregator $A_k$ with its $ID_k$ and $HT_{ID_k}$;

9.     **Else:**

10.       upload only local dataset hash $HCD_k$ and FL client program $CP$ to

11.       local aggregator $A_k$ with its $ID_k$ and $HT_{ID_k}$;

12. **Client Verification:** $A_k$ verify client $k$ FL program and local dataset hash $HCD_k$

13. **For** each client $k$:

14.     **If** client $k$ in untrusted client $UC_k$ list:

15.       **If** client k $HT_{ID_k}$ mismatch

16.         $A_k$ remove client k from FL training; **Break;**

17.       **If** client $k$ upload $\mathbb{X}_k$ to $A_k$:

18.         Generate $HDT_{UC_k}$ delegation token, and new client $ID_{UC_k}$ for $UC_k$ then

19.         delegate the training process to other trusted client $k$;

20.         $A_k$ remove client $k$ from FL training;

21.         **For** valid client $k$:

22.           **If** client $k$ $\tau_k \geq 15$ **And** not been delegated, excluding aggregator:

23.           Send $UC_k$ with $\mathbb{X}_{UC_k}$, $HDT_{UC_k}$ and New $ID_k$. Then client $k$ start another

24.           client instance using $\mathbb{X}_{UC_k}$ as dataset, $HDT_{UC_k}$ as token and $ID_{UC_k}$. **Break;**

25.         **End For**

26.         **If** no trusted client is available: $A_k$ perform the training for $UC_k$;

27.       **Else:** $A_k$ remove client $k$ from FL training;

28.     **Elif** client $k$ **Not** in untrusted client $UC_k$ list:

29.       **If** client k $HT_{ID_k}$ **Or** $HCD_k$ **Or** $HCP$ mismatch with information received from $GS$:

30.         $A_k$ remove client k from FL training;

31.     $A_k$ update TPM latest $\tau_{D_i}$, $HDT_{UC_k}$ and other hashes;

32.     $\{HAT_{Dk}, \tau_{D_i}, HCP, HACP, HT_{ID_i}, HCD_{ID_i}, HDT_{UC_k}\} \rightarrow TPM(A_k)$ for $ID_i$;

33. **End For**

34. **Domain-Level Aggregation:** Send all client $k$ $W_{r=0}$ and hyperparameters;

35. **For** each round $r$ do:

36.     **For** each client $k$ do:

37.       Send updated weight to the client $k$: $W_r^k = W$; Client $k$ trains the model on

---

| Algorithm 2 (continued) | |
|---|---|
| 38. | its dataset $\mathbb{X}_k$ using $W_r^k$ *and* generate local model parameters; |
| 39. | $W_{r+1}^k = \operatorname{argmin}(W_r^k, \text{loss function}(W_r^k))$; Client $k$ send $W_{r+1}^k$ with $HT_{Dk}$; |
| 40. | **End For** |
| 41. | $A_k$ aggregates client $k$ weight $W_r^k$ with **Adaptive Aggregation**$(W_r^k)$, |
| 42. | $\tilde{f}_{Domain}(.) = $ **Adaptive Aggregation**$(W_r^k)$; |
| 43. **End For** | |
| 44. **Global-Level Aggregation:** | |
| 45. **For** each $A_k$ for all $D_k$ do: | |
| 46. | Send $HAT_{Dk}$ and $\tilde{f}_{Domain}(.)$ weight $W_{r+1}^D$ to global aggregator server; |
| 47. | **If** $HAT_{Dk}$ valid: Accept $W_{r+1}^D$; **Else:** Drop $W_{r+1}^D$; |
| 48. **End For** | |
| 49. The global server aggregates the domain model $W_{r+1}^D$ and compute weighted average, | |
| 50. then distributes the weighted average to all $A_k$; $W_{r+1}^k \rightarrow all\ A_k$. After that $A_k$ | |
| 51. updates the weight of $\tilde{f}_{Domain}(.)$ to form a global model $\tilde{f}_{Global}(.)$. Then $A_k$ | |
| 52. distribute updated $\tilde{f}_{Global}(.)$ to all client $k$ in $D_k$. $\tilde{f}_{Global}(.) \rightarrow all\ k$; | |

values mismatch with information received from *GS*, $A_k$ Remove the client from FL training and its data as it indicates the client has malicious intentions. Before completing the verification process, $A_k$ once again updates its TPM to include changes related to token and trust score.

Once the domain-level verification is done, FL training starts when the local aggregator $A_k$ sends the clients in domain $D_k$ with the initial NTC model weight $W_{r=0}$ and training hyperparameters. This is to ensure all local clients are training using the same model structure and parameters, such as r training round, $E$ epochs, $\eta$ learning rate, and $\beta$ batch size. The clients train the NTC model using its local dataset and then send its weight parameters along with a verification token. If the token is valid, the local aggregator aggregates the local model's weight parameters *via* the adaptive aggregation mechanism. At the end of the training round, the local aggregator produces a domain model $\tilde{f}_{Domain}(.)$ based on parameter updates from the local clients. The adaptive aggregation process will be discussed in the following subsection.

After training the domain model, the local aggregator in each domain sends the weight of the domain model to the global aggregator server along with the aggregator token $HAT_{Dk}$ for verification. If the token is valid, the global server aggregates the domain model's weights using averaging methods and distributes back the weighted average to all local aggregators. After that, the local aggregator updates the weight of the domain model to form a global NTC model $\tilde{f}_{Global}(.)$ which is then distributed to all clients for inferences and classifying traffic.

(c) *Adaptive aggregation.* During hierarchical training, the framework aggregates the weight parameters of the clients *via* an adaptive aggregation process, as detailed in

---

**Algorithm 3** Adaptive aggregation process.

---

**Require:** $\tau_{D_i}$ for $D_k$, $W_{r=0}$, r training round, $E$ epochs, $\eta$ learning rate, $\beta$ batch size, $Gt = 2$ GAN threshold.

1. **Check for GAN Attack:**

2. **If** round $r = 1$:

3.     **For** each client $k$ do:

4.         Evaluates client $k$ parameter update as $\tilde{f}_{Eval}(.)$. class_index = 0;

5.         **For** class_F1_Score in $\tilde{f}_{Eval}(.)$ do:

6.             **If** client $ID_k == ID_{A_k}$: eval_bench[class_index] = class_F1_score;

7.             **Else:**

8.                 **If** eval_bench[class_index] >= 0.1:

9.                     **If** class_F1_Score < 0.1: gan_count++; gan_client.append($ID_k$);

10.             class_index ++;

11.         **End For**

12.     **End For.** **If** gan_count >= $Gt$: ganStatus = **True;**

13. **Calculate Client Weightage:** Find $\tau_{Median}$ median value of client trust score.

14. **If** ganStatus == **True:** count(gan_client) as $h$ and $k$ – count(gan_client) as $K$;

15.     **For** each client $k$ do:

16.         **If** client $k$ in gan_client: Assign client weight $CW_k = \dfrac{10}{h}$; **Else:** $CW_k = \dfrac{90}{K}$;

17.     **End For**

18. **Else If** all client $k$ $\tau_{D_i} \geq \tau_{Median}$: all client $k$ weight $CW_k = \dfrac{100}{k}$;

19. **Else:** Find number of clients $k$ where $\tau_k \geq \tau_{Median}$ as $K$ and $\tau_k < \tau_{Median}$ as $h$;

20.     **For** each client $k$ do:

21.         **If** client $k$ $\tau_k \geq \tau_{Median}$: Assign client weight $CW_k = \dfrac{80}{K}$; **Else** $CW_k = \dfrac{20}{h}$;

22.     **End For**

23. **Model Aggregation:**

24. **For** each round $r$ do:

25.     **If** round $r = 1$: $AGG_{Best}() = FedAvg(W_{r+1}^k)$;

26.     **Else If** round $r = 2$:

27.         $A_k$ aggregates client $k$ local model weight $W_{r+1}^k$ *via* multiple algorithms to

28.         form multiple test models $\tilde{f}Test_i(.)$; *e.g.*, Median Mean, Krum;

29.         $A_k$ evaluate all $\tilde{f}Test_i(.)$ using $\mathbb{X}_T$ and find the highest value;

30.         $F1_i = Evaluate\left(\tilde{f}Test_i(.), \mathbb{X}_T\right)$ where $i = MM, TM10, \dots, k$ algorithm;

31.         $F1_{max} = \max(F1_1, F1_2, .., F1_k)$;

32.         $\tilde{f}Test_k(.)$ with the **highest** accuracy selected as the best algorithm $AGG_{Best}()$;

33.     **Else:** $AGG_{Best}() = AGG_{Best}(W_{r+1}^k)$;

34.     $A_k$ aggregates client $k$ local model weight $W_{r+1}^k$ with $AGG_{Best}()$ and set weight

35.     of each client $k$ based on $CW_k$; **Return** $W_{r+1}^k = AGG_{Best}(W_r^k) * CW_k$;

36. **End For**

---

 

Algorithm 3. It starts with checking the parameters update for a potential GAN attack. During the initial training round, the local aggregator $A_k$ forms an evaluation model $\tilde{f}_{Eval}(.)$ for each client $k$ including the local aggregator from the parameter update. The F1-score for each class in the local aggregator evaluation model becomes a benchmark for evaluating the possibility of a GAN attack in the clients' parameter update. When evaluating the clients' update, if it causes the number of classes with F1-scores less than 0.1 or 10% to exceed the threshold, it indicates that the client is attempting a GAN attack. In this study, the GAN attack threshold is set to two classes.

After checking for GAN attack, the local aggregator calculates the clients' weightage $CW_k$ based on three scenarios. First, if there is an attempt to send a poisonous GAN attack *via* an update, 10% of the weightage is divided among the suspected clients, while 90% is divided among the trusted clients. This limits the influences of the GAN attack while ensuring a portion of information from the malicious clients is learned. The second scenario is when there is no suspected GAN attack and the trust score $\tau_k$ of all clients equal to or more than the median value. This indicates an acceptable trust level among all clients and thus the weightage is divided equally among the clients so that each update has equal influence in the domain model $\tilde{f}_{Domain}(.)$. The third scenario is when there is no suspected GAN attack; however, some of the clients' trust scores are below the median level. Thus, 20% of the weightage is divided among the risky clients, while 80% of the weightage is divided among the trusted clients to limit the influence of the risky clients.

After calculating and assigning weight to each client, the local aggregator begins aggregating the clients' parameter updates. For the first round, aggregation is done *via* the standard FedAvg algorithm because, in the first round, the local model has not incorporated the weight $W_{r+1}^k$ learned from other clients within the domain $D_k$. However, in the second round, the local aggregator aggregates the clients' parameters using multiple algorithms to form multiple test model $\tilde{f}Test_i(.)$. The algorithm includes FedAvg, Weighted averaging, Krum, Multi-Krum, Median, and Trim Mean. The local aggregator evaluates the performance of each test model and algorithm with the highest accuracy is selected as the best aggregation algorithm $AGG_{Best}()$. This is because some algorithms are more effective in certain conditions or attacks. For example, FedAvg provides the best performance in normal conditions, while a distance-based approach such as Krum is more effective for LF attacks.

Moreover, during aggregation, the local aggregator also set the weight for each client's parameters based on the assigned weight $CW_k$ calculated earlier to increase or decrease the clients' influences. For the following training round, the local aggregator uses the selected best algorithms for aggregation. At the end of each training round, the local aggregator returns the updated weight $W_{r+1}^k$ back to the clients.

## Federated learning testbed setup

The FL testbed consists of a single aggregation server and six edge clients; each of the hosts was assigned a different TCP/IP port number. Each of the edge clients trains a local model

using its local dataset $\mathbb{X}_k$ and sends the model parameters $W_r^k$ to the aggregation server to form a global model $\tilde{f}_{Global}(.)$. The client trains a DL-based NTC model, which involves processing the raw network packet byes into labeled packet byte matrix (PBM) (*Wang et al., 2018*). This study uses MLP with two fully connected hidden layers, each with six nodes, and the rectified linear unit (ReLU) activation function. The input layers accept input with 740 packet bytes as features and use ReLU as the activation function. Meanwhile, the output layer classifies traffic into ten classes and uses Softmax as the activation function.

Besides MLP, we also used 1D-CNN and 2D-CNN architectures to train the local models. This ensures a fair benchmark against existing work by *Sameera et al. (2024)* and *Thein, Shiraishi & Morii (2024)*. This study CNN architecture begins with a convolutional layer containing 64 filters, followed by a max-pooling layer, and then a second convolutional layer with 128 filters. Both convolutional layers use a kernel size of 3 and the ReLU activation function. After the convolutions, we apply a flatten layer before a fully connected layer of 128 nodes, which also uses ReLU activation. To prevent overfitting, we include a dropout layer with a rate of 0.5 just before the final output. Since CNNs require input data in a specific pixel-like format, we reshaped the datasets accordingly. For example, we reshaped the Fashion-MNIST dataset into $28 \times 28 \times 1$ arrays and the CIFAR-10 dataset into $32 \times 32 \times 3$ arrays.

The model uses categorical cross-entropy as the loss function and ADAM as the training optimizer. The training learning rate is set to $\eta = 0.001$, batch size is $\beta = 64$, with FL training round r = 3 and each round epoch $E = 36$. During the experiment, several aggregation algorithms were used as a benchmark for the defensive effectiveness of the proposed framework. The FedAvg is the default FL algorithm and does not have any defense against adversarial attacks. Meanwhile, other robust aggregation algorithms include weighted averaging (WA), median mean (MM), trimmed mean (TM) with a 10% trim rate, Krum, and Multi-Krum.

For the implementation of the framework and experiment, this study uses Python 3.8, including libraries such as scikit-learn 1.5.1, PyShark 0.3.6, TensorFlow 2.12.1, CUDNN 8.9, Twisted 18.9.0, Flower 1.6.0 (*Beutel et al., 2020*) and WandB. The source code is made available on the GitHub (*Ariffin, 2025*). The experiment was conducted on a host with an AMD Ryzen 7 7840HS 8-core CPU, 16 GB DDR5 Memory, and an Nvidia RTX 4070 GPU. The host runs Ubuntu 20.04 LTS for its OS and other programs such as CUDA 12.6, IBM TPM 2.0 simulator, and TPM 2.0 libraries such as TSS 3.1.0, ABRMD 2.3.1, TSS-engine 1.1.0, Tools 4.3.2, and OpenSSL 1.1.2.

## Datasets and preprocessing

This study uses the ISCX-VPN 2016 network traffic dataset (*Draper-Gil et al., 2016*) which contains packet capture of popular network protocols or services, including encrypted SSL/TLS traffic. This study selected ten traffic classes, which are: (0) AIM Chat, (1) Email, (2) Facebook Audio, (3) Facebook Chat, (4) Gmail Chat, (5) Hangouts Chat, (6) ICQ Chat, (7) Netflix, (8) Spotify, (9) YouTube. The total number of traffic packet instances selected is 671,326, and after randomly splitting training and evaluation datasets at a ratio of 70:30,

**Table 4 Details on the ISCX-VPN 2016 dataset.**

| Class No | Protocol/ Services | Number of instances | Description |
|---|---|---|---|
| 0 | AIM chat | 4,946 | Popular instant messengers developed by AOL allow users to send messages and files to each other. |
| 1 | Email | 47,568 | It is generated *via* Thunderbird client utilising SMTP/S, POP3/SSL, and IMAP/SSL protocol. |
| 2 | Facebook audio | 275,156 | Voice over IP service provided by Facebook. |
| 3 | Facebook chat | 16,104 | Instant messaging features are provided by Facebook. |
| 4 | Gmail chat | 24,172 | Formerly known as Google Talk, it is an Instant messaging feature integrated within Google's email service. |
| 5 | Hangouts chat | 20,016 | Google's unified communication services integrate services like Google Talk, Google+ Messenger, and the original Hangouts video chat service. |
| 6 | ICQ chat | 4,662 | It was one of the earliest instant messaging platforms on the Internet, and it allowed users to send messages, files, voice messages, and video chats in real time. |
| 7 | Netflix | 207,442 | A popular streaming service that offers a wide variety of TV shows and movies on thousands of internet-connected devices. Use SSL/TLS to secure streaming traffic. |
| 8 | Spotify | 30,034 | Popular digital music streaming service. Use HTTPS/SSL to secure traffic. |
| 9 | YouTube | 41,226 | Popular video sharing and social media platform that allows users to upload and share videos. Use HTTPS/SSL to secure traffic. |

the training instance becomes 469,928, and the evaluation instance becomes 201,398. Table 4 provides details about the ISCX-VPN 2016 dataset, including a breakdown of class instances and descriptions of the traffic data. This breakdown indicates an imbalanced distribution of traffic classes, a characteristic that researchers often find in network traffic data (*Abdelkhalek & Mashaly, 2023*). This study maintains this imbalanced distribution in the training datasets to simulate non-IID (non-independently and identically distributed) conditions when evaluating the FL solutions (*Zhang et al., 2021b*). Researchers often refer to this specific type of statistical heterogeneity as label distribution skew or class imbalance. It is a primary attribute of non-IID data, especially in FL environments where clients collect data from different network environments (*Jimenez G et al., 2024*).

The raw packet data in the ISCX-VPN 2016 dataset must undergo preprocessing before it can be used for FL training. The first step of the preprocessing is parsing the packet data to remove the data link layer or Ethernet header, as it only contains significant local information. Moreover, we also remove the source and destination IP address as the information is constantly changing and only significant for specific network environments. After that, the raw packet bytes undergo a padding or truncating process to limit or make the size of each instance within 740 bytes or half the size of the maximum transmission unit (MTU).

Then the raw bytes are transformed into packet byte vectors (PBV) $X_i = \{x_{i1}, x_{i2} \ldots x_{ij}\}$, where $i$ represents the dataset and $j$ represents the j-th byte in $X_i$. Each PBV needs to be associated with a traffic label $y$ (*e.g.*, Email, Facebook, YouTube), with each byte in the PBV serving as an input feature. After that, we aggregate all PBV together with its labels to the form of $\mathbb{X} = \{X_1^T, X_2^T \ldots X_i^T\}^T$, where $i$ is the number of PBV

**Table 5 Details on the N-BaIoT dataset.**

| Traffic type | Attack class | Number of instances |
|---|---|---|
| Mirai | Scan | 7,000 |
| | UDP | 7,000 |
| | UDP plain | 7,000 |
| | Syn | 7,000 |
| | Ack | 7,000 |
| BashLite | Scan | 9,000 |
| | Junk | 9,000 |
| | UDP | 9,000 |
| | TCP | 9,000 |
| | Combo | 9,000 |
| Benign | – | 90,000 |

datasets. Then we combine all processed PBV into a packet bytes matrix (PBM), where each row represents packet instances, and each column represents byte features along with its label; as such $\mathbb{X} = \begin{bmatrix} X_1 \\ \ldots \\ X_i \end{bmatrix} \leftarrow \begin{bmatrix} y_1 \\ \ldots \\ y_i \end{bmatrix}$, where $y_i$ is the traffic label. The label is then converted into a one-hot encoding format. After converting to PBM format, we normalize the values *via* the L2 normalization method along the y-axis as the DL model achieved faster convergence and higher performance with normalized value. The last steps of pre-processing involve dividing both training and evaluation datasets into six data shards, $\mathbb{X}_1, \mathbb{X}_2 \ldots \mathbb{X}_k$ which are used by the edge clients as their local datasets. Each of the clients' data shards maintains a skewed distribution of traffic classes to create non-IID characteristics. In addition to ISCX-VPN 2016, this study uses other public datasets, such as N-BaIoT, Fashion-MNIST, and CIFAR-10. The N-BaIoT dataset (*Meidan et al., 2018*) contains real network traffic data that researchers collected from multiple commercial IoT devices infected with popular botnet malware. Table 5 provides details regarding the data instances of the N-BaIoT datasets. This study uses the N-BaIoT dataset to provide a fair comparison with the reported value in the work of *Thein, Shiraishi & Morii (2024)*; therefore, we preprocessed the data following the methodology they described in their work. Meanwhile, the Fashion-MNIST dataset (*Xiao, Rasul & Vollgraf, 2017*) contains 70,000 instances of grayscale (single-channel) images, and each image is a 28 × 28 matrix of pixel intensities. The dataset has 10 classes that represent different types of clothing and fashion accessories, such as T-shirts/tops, Trousers, and Sneakers. This study uses this dataset to provide a fair comparison with the reported value in the work of *Sameera et al. (2024)*.

Lastly, the CIFAR-10 dataset (*Krizhevsky, 2009*) contains 60,000 instances of RGB (3-channel) color images. Each image is a 32 × 32 matrix with pixel values that range from 0 to 255. The dataset has 10 classes that represent common objects from various categories, including animals, vehicles, and transportation. This study uses this dataset to provide a fair comparison with reported value in the work of *Cao et al. (2024)*. For our experiments,

we divide the N-BaIoT, Fashion-MNIST, and CIFAR-10 datasets into six training and evaluation data shards for the clients.

## Attack scenarios

During the evaluation, this study simulates different adversarial attacks while training the NTC model *via* FL. This study conducts multiple experiments for each attack type. The experiment starts with a normal scenario with no attacker, which serves as a control or baseline. Then, we repeat the experiment with one, two, and up to four attackers to simulate a collusive environment. This study simulates the following four types of adversarial attacks:

(a) *Label flipping attack.* This study simulates two variants, which are all and class label flipping (LF) attacks. Both attacks involve tampering with the local datasets of participating FL clients. The all-LF involves flipping the label of all classes. Where $\mathbb{X}_i$ has a true label $y_i$ for class $C_i, y_i \in \{C_1, C_2, \dots, C_n$ and flipping function as $f: y_i \rightarrow y'_i$. The attack aims to degrade overall classifier performance. Meanwhile, the class-LF involves flipping only the label of FB-audio (class no 2) traffic to degrade the class performance. Where in $\mathbb{X}_i$ if it's $C_{FBAudio}$ flips to a random label $C_i$, otherwise, the label remains unchanged $f(y_i) = \begin{cases} C_i \text{ if } y_i = C_{FBAudio} \\ y_i \text{ otherwise} \end{cases}$.

(b) *Model poisoning attack.* The attacks involve tampering with clients' FL programs to manipulate the model structure and the values in the parameters update. This study simulates two variants of the attack: model cancelling and gradient factor attack. To conduct the model cancelling attack, the malicious client sets the model's weight to zero, $\Delta W_m = 0$ (*Rey et al., 2022*). Meanwhile, for the gradient factor attack, we multiply the client gradient with a negative factor $\alpha$, where $\alpha < 0$ (*Blanchard et al., 2017*). Both attacks severely degrade the overall performance of the global model.

(c) *Backdoor attack.* The attack is based on the work of *Bagdasaryan et al. (2018)* which involves injecting backdoor patterns during FL training with the aim of poisoning and causing the global model to misclassify network traffic during inference without affecting the overall accuracy; thus, it is a more targeted attack. For example, when using the ISCX-VPN 2016 datasets, Email traffic (Class No. 1) is selected as the target class $Y_{Target} = 1$, and FB audio (Class No. 2) as the malicious class $M_{Malicious} = \mathbb{X}_k[:, 2]$. Then, this study sets the poison sample number $m = 20,000$ which is used by malicious clients. After that, the backdoor script creates a backdoor pattern $X_B$ for 740 features based on $m$ values, values $= c/m$ where $c$ is the current feature iteration, which starts at zero and increments until it reaches 740.

The script will produce a pattern $X_B$ with 740 columns. Then, during training, the malicious clients randomly select $m$ data instances from their dataset shard and replace the values of the selected instances with the generated backdoor pattern. The malicious clients also assign the backdoor data instances' label as target class label, $Y_k[Y_{Random}, Y_{Target}] = 1$. During the evaluation, this study injects the same backdoor pattern $X_B$ into the FB Audio instance to make the model misclassify the traffic as Email traffic. The attack poison rate is calculated as $P_{rate} = \dfrac{m}{Total(\mathbb{X}_k)} \times 100$, where

$Total(\mathbb{X}_k)$ is the total number of instances in the training dataset. Therefore, in a scenario with one attacker, the rate is calculated as $P_{rate} = \dfrac{20{,}000}{469{,}928} \times 100 = 4.25\%$.

The poisoning rate doubles as the number of attackers increases, since each attacker uses the same poison sample value of $m = 20{,}000$.

(d) **GAN-based attack.** This targeted attack involves generating synthetic data using a generative adversarial network (GAN) model for specific traffic classes and injecting it during FL training to cause a bias during classification (*Zhang et al., 2019*). Before conducting a GAN attack, it is necessary to train both the generator and discriminator models. The generator's input layer takes random noise with a shape of 200, while the output layer synthesizes 740 features mirroring real traffic data and uses the sigmoid activation function. Meanwhile, the discriminator's input layer accepts synthetic data from the generator and real data samples with a shape set to 740. The discriminator model aims to distinguish between real and synthetic data and then calculate the losses for both the generator and discriminator, guiding the improvement of their gradients during training. In this study, we use the trained generator model to synthesize FB Audio traffic and inject the synthetic data into Netflix as target class during training to make the global model bias to classify Netflix (Class no 7) as FB Audio (Class no 2) traffic. The malicious edge client replaces the dataset $\mathbb{X}_k$ target class data in local dataset with the generated synthetic data, $\mathbb{X}_k[Y_{Target}] = x_{Synth}$. Similar to backdoor attacks, for the GAN-based attack, each malicious client sets the poison sample number $m = 20{,}000$. The GAN poison rate increases as the number of attackers increases, $P_{rate} = \dfrac{m}{Total(\mathbb{X}_k)} \times 100$, where $Total(\mathbb{X}_k)$ represents the total number of training datasets from all malicious clients.

## RESULTS AND DISCUSSION

This section discusses the evaluation metrics and results from the FL-based NTC training, which used the SHeRAA-FL framework under various adversarial attacks like LF, MP, backdoor, and GAN-based attacks. We benchmark the framework's effectiveness against existing defensive measures, including WA, MM, TM, Krum, and Multi-Krum. For specific datasets, we include additional comparisons with other defenses: FoolsGold and LFGuard for Fashion-MNIST, pFL-IDS for N-BaIoT, and SRFL for CIFAR-10. Finally, this section discusses the computational overhead of implementing the proposed secure framework.

### Evaluation metrics

This study uses overall accuracy, F1-score, and attack success rate (ASR) metrics to evaluate the performance of the NTC model and the effectiveness of the SHeRAA-FL framework in mitigating adversarial attacks. Both accuracy and F1-score metrics provide statistical validation for evaluating model performance in classifying multi-class network traffic, including imbalanced data scenarios. For LF, model cancelling, gradient factor, backdoor, and GAN-based attacks, we used the overall accuracy metric to measure the

attack's impact on overall model performance. We evaluate the defensive approach's effectiveness by measuring its ability to maintain overall accuracy even when it receives poisonous updates from multiple attackers. Flipped labels, modified gradients, or backdoor patterns cause the client update to become poisonous by misleading its weight.

Meanwhile, for class-LF and GAN-based attacks, we also used the F1-score to assess the attack's impact on specific traffic classes, as that is the attack's aim. An effective defensive approach must prevent a reduction in a specific class's F1-score that results from misleading labels or synthetic data injection. Previous studies have used both overall accuracy and F1-score metrics as common benchmarks to evaluate the effectiveness of defensive measures (*Cao et al., 2024*; *Sameera et al., 2024*; *Rey et al., 2022*; *Thein, Shiraishi & Morii, 2024*; *Zhang et al., 2019*).

Lastly, for the backdoor attack evaluation, this study uses the ASR in addition to overall accuracy. We calculate the ASR based on the number of successful backdoor attacks out of five attempts during inference. We record a backdoor attack as successful when the attacker misclassifies a target class (*e.g.*, FB Audio to email) by injecting the backdoor pattern during inference. In each attempt, the model completes three FL training rounds r = 3. An effective defensive approach must prevent the FL model from misclassifying a class when an attacker injects a backdoor pattern during FL training. A previous study also used the attack success rate metric to evaluate defenses against backdoor attacks (*Cao et al., 2024*). This study also measures the FL training time, maximum CPU, and memory utilization as metrics for evaluating the proposed framework's computational overhead.

## Label flipping experiment

For the LF attack experiment, this study conducted two variations: all-LF and class-LF attacks. Table 6 presents the overall global model accuracy for the all-LF experiment using the ISCX-VPN 2016, Fashion-MNIST, and N-BaIoT datasets. Table 7 presents the F1-score of the target class (FB Audio) for the class-LF experiment using the ISCX-VPN 2016 dataset.

The all-LF experiment results in Table 6 show that under normal circumstances with the ISCX-VPN 2016 dataset, SHeRAA-FL produces the NTC model with the highest accuracy (0.9131). This indicates that the framework's defensive measures cause minimal disruption to the training process. In adversarial scenarios, the framework consistently produces a model with the highest accuracy, even as the number of attackers increases, which shows its effectiveness in mitigating the impact of multiple adversaries. In a scenario with four attackers, the framework achieves a 0.8644 overall accuracy, representing only a 5.33% reduction from the normal scenario. In contrast, other defensive measures recorded severe accuracy reductions; for example, Krum's accuracy fell by 99.6%.

The evaluation using Fashion-MNIST, shown in Table 6, reveals that SHeRAA-FL provided better protection against the all-LF attack by maintaining a model with higher accuracy. For example, in the all-label LF scenario with three attackers, SHeRAA-FL achieved an accuracy of 0.9287. This accuracy was 2.91% higher than LFGuard, 4.48% higher than FoolsGold, 4.73% higher than Krum, and 4.77% higher than Multi-Krum. The evaluation using N-BaIoT involved a 30% attack ratio in which two clients became

**Table 6 All label flipping attack experiment results.**

| Defensive measures | Overall accuracy | | | | |
|---|---|---|---|---|---|
| | ISCX-VPN 2016 dataset | | | | |
| | Number of attackers/Scenario | | | | |
| | Normal | 1 | 2 | 3 | 4 |
| FedAvg | 0.9042 | 0.7227 | 0.4651 | 0.2618 | 0.0472 |
| WA | 0.9035 | 0.7929 | 0.7726 | 0.7868 | 0.8236 |
| MM | 0.9054 | 0.8934 | 0.8865 | 0.8326 | 0.0406 |
| TM | 0.8701 | 0.8455 | 0.8377 | 0.1486 | 0.0336 |
| Krum | 0.9079 | 0.8971 | 0.8866 | 0.8006 | 0.0036 |
| Multi-Krum | 0.8795 | 0.9143 | 0.8984 | 0.8905 | 0.0037 |
| SHeRAA-FL | 0.9131 | 0.9145 | 0.9100 | 0.8937 | 0.8644 |
| | Fashion-MNIST dataset | | | | |
| | 1 | | 2 | 3 | |
| Krum | 0.8892 | | 0.8872 | 0.8848 | |
| Multi-Krum | 0.8860 | | 0.8857 | 0.8844 | |
| FoolsGold | 0.8900 | | 0.8885 | 0.8871 | |
| LFGurad | 0.9054 | | 0.9041 | 0.9017 | |
| SHeRAA-FL | 0.9279 | | 0.9267 | 0.9287 | |
| | N-BaIoT dataset | | | | |
| | Normal | | All-label flipping (30% Attack ratio) | | |
| | 0.9794 | | 0.4706 | | |
| MM | 0.9859 | | 0.9723 | | |
| TM | 0.9885 | | 0.4706 | | |
| Multi-Krum | 0.9822 | | 0.9598 | | |
| pFL-IDS | 0.9964 | | 0.9951 | | |
| SHeRAA-FL | 0.9794 | | 0.4706 | | |

**Table 7 Class label flipping attack experiment results using the ISCX-VPN 2016 dataset.**

| Defensive measures | Target class F1-score (FB-audio) | | | | |
|---|---|---|---|---|---|
| | Number of attackers | | | | |
| | Normal | 1 | 2 | 3 | 4 |
| FedAvg | 0.9809 | 0.0000 | 0.0000 | 0.0000 | 0.0000 |
| WA | 0.9839 | 0.0000 | 0.0000 | 0.0000 | 0.0000 |
| MM | 0.9773 | 0.9423 | 0.9132 | 0.0000 | 0.0213 |
| TM | 0.9713 | 0.0285 | 0.0000 | 0.0000 | 0.0000 |
| Krum | 0.9793 | 0.9785 | 0.9794 | 0.9639 | 0.2686 |
| Multi-Krum | 0.9773 | 0.9762 | 0.9731 | 0.9696 | 0.3217 |
| SHeRAA-FL | 0.9826 | 0.9701 | 0.9709 | 0.8639 | 0.9550 |

**Table 8  Model cancelling attack experiment results using the ISCX-VPN 2016 dataset.**

| Defensive measures | Overall accuracy | | | | |
|---|---|---|---|---|---|
| | Number of attackers | | | | |
| | Normal | 1 | 2 | 3 | 4 |
| FedAvg | 0.9042 | 0.7790 | 0.4015 | 0.3602 | 0.4623 |
| WA | 0.9035 | 0.8632 | 0.8662 | 0.6931 | 0.8528 |
| MM | 0.9054 | 0.9047 | 0.8927 | 0.7360 | 0.4082 |
| TM | 0.8701 | 0.8748 | 0.4411 | 0.5304 | 0.3801 |
| Krum | 0.9079 | 0.9131 | 0.9065 | 0.4082 | 0.4082 |
| Multi-Krum | 0.8795 | 0.8997 | 0.8655 | 0.4082 | 0.4082 |
| SHeRAA-FL | 0.9131 | 0.9028 | 0.9098 | 0.9069 | 0.9028 |

malicious. The results showed a similar trend where SHeRAA-FL provided better protection. Under normal conditions, SHeRAA-FL produced a model with the highest accuracy of 0.9964. Under adversarial conditions, SHeRAA-FL maintained the highest accuracy of 0.9951, a reduction of only 0.13%. In contrast, pFL-IDS reported a 2.28% reduction, while Multi-Krum experienced a 52.39% reduction.

On the other hand, the class-LF experiment results in Table 7 show that in normal circumstances, the SHeRAA-FL framework achieved a 0.9826 F1-score for the FB Audio class, slightly behind WA (0.9839). In the class-LF attack scenario with one and two attackers, Krum produced the model with the highest F1-score (0.9785 and 0.9794, respectively), while our framework achieved 0.9701 and 0.9709. With three attackers, Multi-Krum had the highest F1-score (0.9696), while our framework obtained 0.8639. However, with four attackers, both Krum's and Multi-Krum's performance started to degrade. At the same time, our framework obtained the highest F1-score of 0.9550, a reduction of only 2.8% compared to the normal scenario.

These results demonstrate that SHeRAA-FL consistently mitigates the impact of the LF attack, even in collusive scenarios. In contrast, some defensive measures, such as WA and TM, produce models with a zero F1-score for the FB Audio class. The results from both LF attack variants show that the mechanisms in SHeRAA-FL effectively mitigate and recover from the attack's impact during training. The remote attestation scoring mechanism correctly identifies clients' datasets with flipped labels and assigns the trust score accordingly. The hierarchical training then limits the poisonous update within the domain model, and the adaptive aggregation adjusts the weight contribution to minimize the update's impact on the global model.

## Model poisoning experiment

For the MP attack experiment, this study conducted two variations: model canceling and gradient factor attacks. Table 8 presents the overall model accuracy under a model canceling attack using the ISCX-VPN 2016 Dataset. Table 9 presents the results for the gradient factor attack using the ISCX-VPN 2016 and N-BaIoT datasets.

**Table 9 Gradient factor attack experiment results.**

| Defensive measures | Overall accuracy | | | | |
|---|---|---|---|---|---|
| | ISCX-VPN 2016 dataset | | | | |
| | Number of attackers/Scenario | | | | |
| | Normal | 1 | 2 | 3 | 4 |
| FedAvg | 0.9042 | 0.0713 | 0.0713 | 0.4082 | 0.3097 |
| WA | 0.9035 | 0.0713 | 0.4082 | 0.4082 | 0.3097 |
| MM | 0.9054 | 0.8474 | 0.7713 | 0.3097 | 0.4082 |
| TM | 0.8701 | 0.0616 | 0.0713 | 0.3097 | 0.4082 |
| Krum | 0.9079 | 0.9084 | 0.8877 | 0.8859 | 0.9072 |
| Multi-Krum | 0.8795 | 0.9118 | 0.8818 | 0.8515 | 0.4082 |
| SHeRAA-FL | 0.9131 | 0.8993 | 0.8488 | 0.9063 | 0.9123 |
| | N-BaIoT dataset | | | | |
| | Normal | Gradient factor (30% Attack ratio) | | | |
| MM | 0.9794 | 0.4706 | | | |
| TM | 0.9859 | 0.4706 | | | |
| Multi-Krum | 0.9885 | 0.4706 | | | |
| pFL-IDS | 0.9822 | 0.9642 | | | |
| SHeRAA-FL | 0.9964 | 0.9942 | | | |

The results of the model canceling attack in Table 8 show that the SHeRAA-FL framework produces a model with the highest accuracy in normal and most adversarial scenarios. The exception is the one-attacker scenario, where Krum obtained an accuracy of 0.9131, while our framework obtained 0.9028. In a scenario with four attackers, the framework achieves an overall accuracy of 0.9028, a reduction of only 1.13% from the normal scenario. In contrast, the second-most effective measure is WA, with an accuracy of 0.8528, a 5.61% reduction. Other defensive measures recorded a severe reduction in accuracy; for example, the TM measure recorded a 56.32% reduction.

Table 9 presents the gradient factor attack results. In the evaluation using the ISCX-VPN 2016 dataset, the Multi-Krum and Krum measures obtained the highest accuracy in the one-attacker (0.9118) and two-attacker (0.8877) scenarios, while our framework achieved 0.8993 and 0.8488 accuracy, respectively. However, in the normal, three-, and four-attacker scenarios, SHeRAA-FL obtained the highest model accuracy compared to other measures. For instance, with four attackers, the framework produces a model with 0.9133 accuracy, only a 0.08% reduction from the normal scenario. The second most effective measure is Krum, with an accuracy of 0.9072.

For the evaluation using the N-BaIoT dataset involved a 30% attack ratio where two clients became malicious. The results showed that SHeRAA-FL provided better protection against the gradient-factor attack compared to existing approaches. Under normal conditions, SHeRAA-FL produced a model with the highest accuracy of 0.9964. Under adversarial conditions, it maintained the highest accuracy of 0.9942, a reduction of only

0.22%. In contrast, pFL-IDS reported a 1.83% reduction, while Multi-Krum showed a 52.39% reduction.

The results from Tables 8 and 9 show that SHeRAA-FL consistently and effectively mitigates the effects of model poisoning attacks, even in collusive scenarios. The results indicate that the framework's mechanisms, when working in tandem, effectively mitigate the impact of these attacks. During remote attestation, the framework correctly identifies clients with tampered FL programs or model structures. Then, during hierarchical training, the framework delegates the training of malicious clients to other clients to minimize the attack's impact. Finally, the adaptive aggregation mechanism adjusts the weight for each client to cater to the delegation process and chooses the best aggregation algorithm.

## Backdoor attack experiment

This section discusses the backdoor attack experiment. Table 10 presents the backdoor experiment results, showing the NTC model's overall accuracy and the attack success rate (ASR) using the ISCX-VPN 2016 and CIFAR-10 datasets.

The evaluation using the ISCX-VPN 2016 dataset shows that a single attacker cannot influence misclassification on the global model. However, with two or three attackers, the backdoor attack achieved a 20–40% success rate when using FedAvg, MM, Krum, or Multi-Krum for aggregation. With four attackers, the attack achieved a success rate of up to 80% for most aggregation approaches, except for SHeRAA-FL. The framework consistently mitigates the backdoor attack and prevents traffic class misclassification. Meanwhile, Table 10 also shows that the backdoor attack did not degrade the overall model accuracy even with multiple attackers, which demonstrates the attack's subtle, class-specific nature. For example, in a scenario with four attackers using Krum, although the attack achieves an 80% success rate, the overall accuracy remains at 0.9002.

The evaluation using the CIFAR-10 dataset involved a 20% poison rate, in which four clients became malicious. The results showed that under normal conditions, SHeRAA-FL obtained a slightly lower overall accuracy (0.7957) than the value SRFL reported (0.8071). We expected this lower performance, as the SRFL study used a more complex 2DCNN neural network structure with multiple 64-node 2DCNN layers and 128-node fully connected layers. However, in adversarial conditions, the results showed that SHeRAA-FL more effectively mitigated backdoor attacks. SHeRAA-FL consistently maintained a 0% attack success rate. In contrast, the SRFL approach recorded a 20% attack success rate, although SHeRAA-FL's overall accuracy was 4% lower. Other defensive approaches performed worse; for instance, MM, TM, and Krum each recorded an 80% ASR.

The backdoor attack experiment results show that SHeRAA-FL effectively mitigates and recovers from the attack. During the attestation process, the mechanism successfully identifies and removes recurring patterns from the poisoned dataset, verifies the data, and minimizes the backdoor's impact on the domain model. Meanwhile, the adaptive aggregation mechanism ensures that the assigned weight for each client reflects its trust score.

**Table 10  Backdoor attack experiment results.**

| Defensive measures | ISCX-VPN 2016 dataset | | | | |
| --- | --- | --- | --- | --- | --- |
| | Number of attackers | | | | |
| | Normal | 1 | 2 | 3 | 4 |
| | Overall accuracy | | | | |
| FedAvg | 0.9042 | 0.9060 | 0.9035 | 0.9101 | 0.9087 |
| WA | 0.9035 | 0.9182 | 0.9082 | 0.8805 | 0.9112 |
| MM | 0.9054 | 0.8801 | 0.8960 | 0.9017 | 0.9099 |
| TM | 0.8701 | 0.8976 | 0.9090 | 0.9114 | 0.9148 |
| Krum | 0.9079 | 0.9086 | 0.9036 | 0.8963 | 0.9002 |
| Multi-Krum | 0.8795 | 0.8910 | 0.8739 | 0.8582 | 0.9016 |
| SHeRAA-FL | 0.9131 | 0.9053 | 0.9171 | 0.9110 | 0.9078 |
| Attack success rate (%) | | | | | |
| FedAvg | – | 0 | 20 | 20 | 60 |
| WA | – | 0 | 0 | 0 | 40 |
| MM | – | 0 | 20 | 40 | 60 |
| TM | – | 0 | 0 | 0 | 40 |
| Krum | – | 0 | 0 | 40 | 80 |
| Multi-Krum | – | 0 | 0 | 20 | 80 |
| SHeRAA-FL | – | 0 | 0 | 0 | 0 |

| | CIFAR-10 dataset | | | |
| --- | --- | --- | --- | --- |
| | Model accuracy | | | |
| | Normal | | Backdoor attack (20% poison rate) | |
| | Overall accuracy | Attack success rate (%) | Overall accuracy | Attack success rate (%) |
| MM | – | – | 0.7856 | 80% |
| TM | – | – | 0.7867 | 80% |
| Krum | – | – | 0.7844 | 80% |
| SRFL (MMRA-MD) | 0.8071 | – | 0.7985 | 20% |
| SHeRAA-FL | 0.7957 | – | 0.7661 | 0% |

## GAN-based attack experiment

This section discusses the GAN-based attack experiment. Table 11 shows the F1-score for Netflix as the target class and FB Audio as the destination class. The results show that in a two-attacker scenario, the WA and TM measures fail to prevent classification bias, as the F1-score of the Netflix class decreases significantly. In a three-attacker scenario, FedAvg, MM, and TM failed to mitigate the attack. In a four-attacker scenario, almost all aggregation measures failed to mitigate the attack except for SHeRAA-FL and WA. For example, with four attackers, our framework maintains an F1-score of 0.9799 for the Netflix target class, a reduction of only 0.44% from the normal scenario. The second most effective measure is WA, with a 0.9059 F1-score, representing a 7.9% reduction. Meanwhile, the F1-score for the Netflix class in other aggregation measures reduces to zero.

**Table 11 GAN-based attack experiment results using ISCX-VPN 2016 dataset.**

| Defensive measures | Number of attackers | | | | |
|---|---|---|---|---|---|
| | Normal | 1 | 2 | 3 | 4 |
| **Target class F1-score (Netflix)** | | | | | |
| FedAvg | 0.9853 | 0.9686 | 0.9412 | 0.6477 | 0.0000 |
| WA | 0.9837 | 0.9768 | 0.5488 | 0.9000 | 0.9059 |
| MM | 0.9848 | 0.9827 | 0.9811 | 0.0519 | 0.0000 |
| TM | 0.9815 | 0.9553 | 0.3173 | 0.6358 | 0.0000 |
| Krum | 0.9832 | 0.9808 | 0.9837 | 0.9795 | 0.0000 |
| Multi-Krum | 0.9812 | 0.9791 | 0.9775 | 0.9777 | 0.0000 |
| SHeRAA-FL | 0.9843 | 0.9831 | 0.9829 | 0.9770 | 0.9799 |
| Destination class F1-score (FB audio) | | | | | |
| FedAvg | 0.9809 | 0.9786 | 0.9768 | 0.9819 | 0.9372 |
| WA | 0.9839 | 0.9754 | 0.9796 | 0.9731 | 0.9724 |
| MM | 0.9773 | 0.9696 | 0.9778 | 0.9814 | 0.9770 |
| TM | 0.9713 | 0.9775 | 0.9763 | 0.9712 | 0.9784 |
| Krum | 0.9793 | 0.9765 | 0.9764 | 0.9709 | 0.9774 |
| Multi-Krum | 0.9773 | 0.9736 | 0.9695 | 0.9730 | 0.9818 |
| SHeRAA-FL | 0.9826 | 0.9759 | 0.9799 | 0.9818 | 0.9758 |

The F1-score for the FB Audio class remained stable, only slightly increasing or decreasing during the attack. This outcome occurs because the GAN-based attack targets only the Netflix class and therefore only slightly impacts other classes. The results indicate that the framework effectively mitigates the GAN-based attack regardless of the number of attackers. The remote attestation process successfully detected the GAN-based attack pattern and assigned a correct trust score to the malicious client. Finally, the adaptive aggregation mechanism minimizes the poisonous update's influence when aggregating the global model.

## Computational overhead

This section discusses the computational overhead of the proposed SHeRAA-FL framework. During the LF attack experiment, this study measures the training time and maximum CPU and memory utilization in normal and multiple-attacker scenarios. We measured these metrics during FL training from one of the local aggregators using the WandB tools. We compare the measured metrics with existing approaches such as FedAvg, WA, MM, TM, Krum, and Multi-Krum.

Figure 5 shows a comparison of training time. In a normal scenario, the comparison shows that SHeRAA-FL incurs an additional 25.28% in training time on average compared to the other approaches. In adversarial scenarios with four attackers, SHeRAA-FL incurs an additional 27.72% in training time on average. We expected this additional overhead because the framework consists of three complex, multi-step mechanisms that work together to mitigate various adversarial attacks.

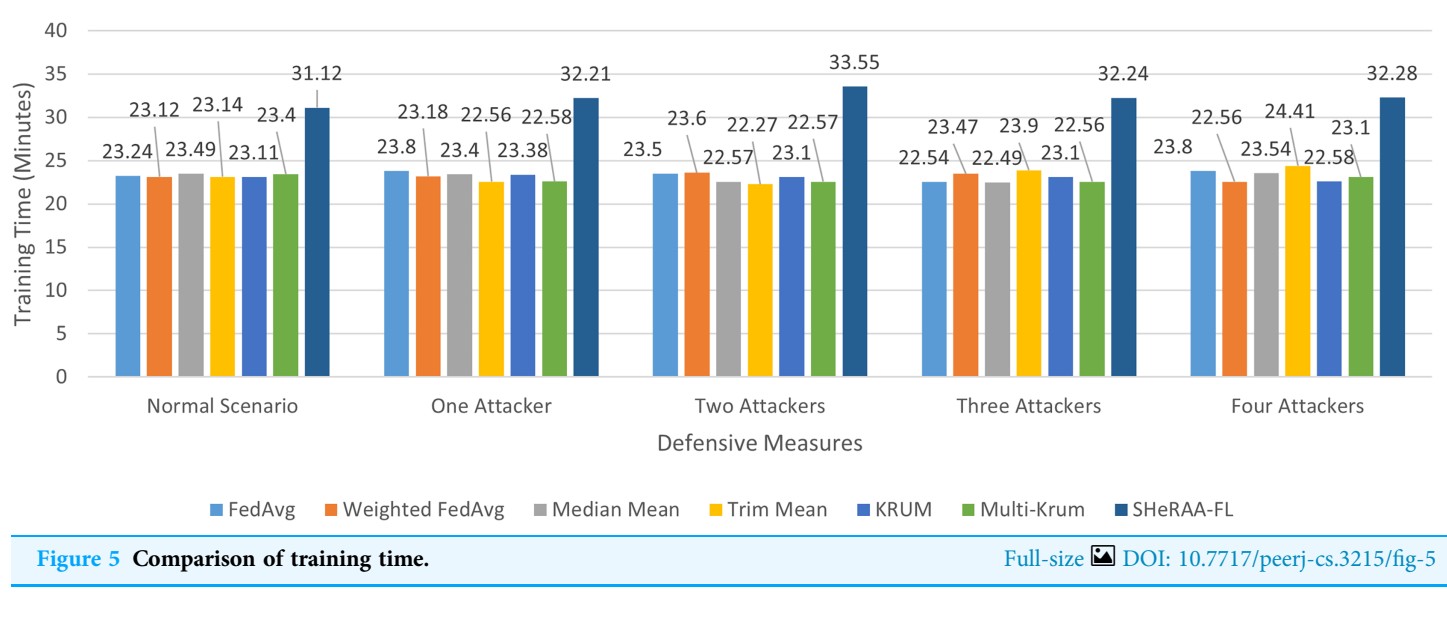

**Figure 5 Comparison of training time.**

| CPU Utilization (%) | Normal Scenario | One Attacker | Two Attackers | Three Attackers | Four Attackers |
|---|---|---|---|---|---|
| ■ FedAvg | 32.36 | 32.55 | 32.84 | 32.84 | 32.87 |
| ■ Weighted FedAvg | 33.11 | 32.47 | 32.89 | 32.39 | 32.86 |
| ■ Median Mean | 32.12 | 32.92 | 32.69 | 32.96 | 31.84 |
| ■ Trim Mean | 32.74 | 32.72 | 33.22 | 32.97 | 32.9 |
| ■ KRUM | 32.37 | 32.55 | 32.77 | 32.49 | 32.97 |
| ■ Multi-Krum | 32.75 | 33.05 | 33.09 | 33.1 | 33.01 |
| ■ SHeRAA-FL | 32.87 | 31.41 | 32.83 | 32.94 | 31.3 |

Defensive Measures

■ FedAvg  ■ Weighted FedAvg  ■ Median Mean  ■ Trim Mean  ■ KRUM  ■ Multi-Krum  ■ SHeRAA-FL

**Figure 6 Comparison of maximum CPU utilization.**

Meanwhile, Fig. 6 compares maximum CPU utilization. The comparison shows that SHeRAA-FL utilizes the same level of CPU time as other approaches. For example, in normal scenarios, the framework and other approaches utilize an average of 32% of CPU time. In adversarial scenarios, the framework utilizes an average of 31–32% of CPU time, which is at the same level as other approaches. The framework consumes the same average CPU time because it processes tasks sequentially and offloads neural network processing to the GPU.

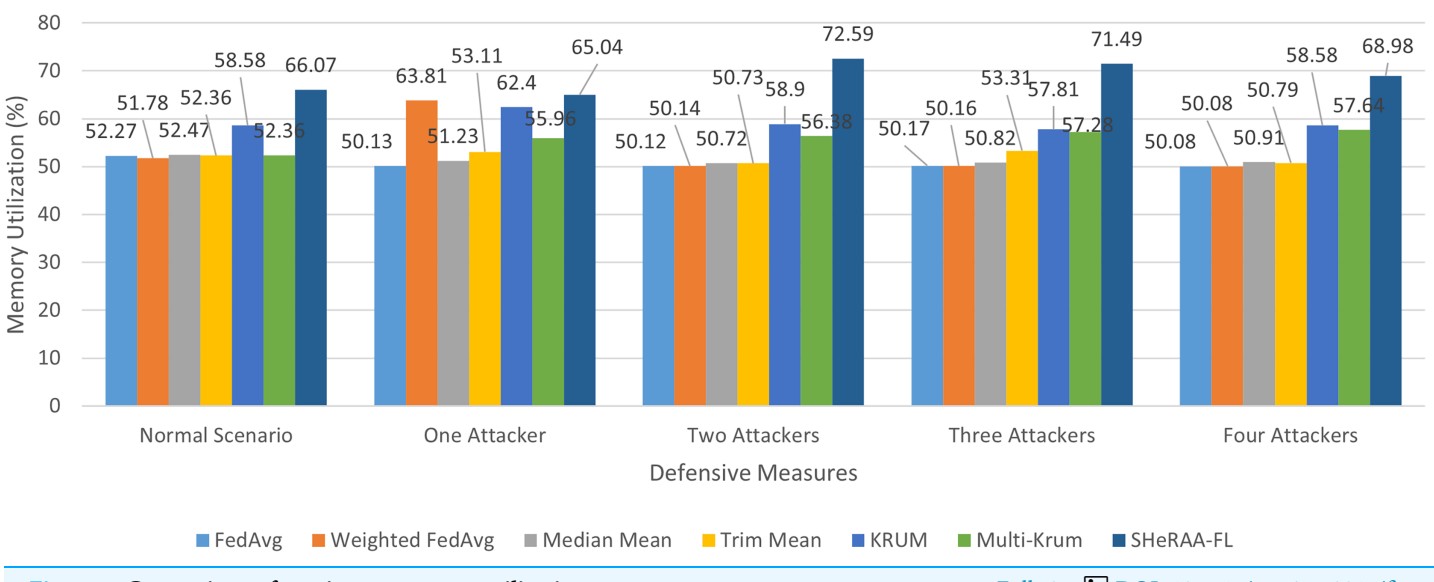

**Figure 7 Comparison of maximum memory utilization.**

Lastly, Fig. 7 compares maximum memory utilization. In a normal scenario, SHeRAA-FL uses 19.32% more memory on average compared to the other methods. In adversarial scenarios, it uses 35.41% more memory on average. This increase occurs because the framework operates the server and clients in a stateful manner, in which the host needs to store additional data such as trust scores and tokens for the verification process. However, despite incurring additional overhead in training time and memory, the framework offers better protection against multiple types of attacks and varying numbers of attackers, as we presented in the previous section. For example, in an all-LF attack scenario with four attackers, SHeRAA-FL maintains a model with 0.8644 overall accuracy, a reduction of only 5.33% from the normal scenario.

## CONCLUSIONS

In conclusion, the distributed nature of FL-based NTC model training makes the model vulnerable to multiple types of adversarial attacks. However, most existing defensive approaches are only effective for certain types of attacks, and their effectiveness diminishes as the number of attackers increases. Moreover, existing defensive measures also lack other security features such as hardware-level security, communication privacy, and identity verification. The lack of such features puts the FL training at risk of tampering and identity-based attacks like Sybil attacks. Therefore, to enhance defenses against multiple types of adversarial attacks, this work proposed the SHeRAA-FL framework. The framework consists of three mechanisms—remote attestation scoring, hierarchical training, and adaptive aggregation—to establish trust among clients and mitigate adversarial impacts during aggregation.

The evaluation results show that the SHeRAA-FL framework effectively mitigates the impact of multiple types of adversarial attacks, such as LF, MP, backdoor, and GAN-based attacks, in a distributed FL environment. Moreover, the framework remains effective even

as the number of attackers increases to four, while other defensive measures fail. The framework also produces a model with the best performance during normal scenarios compared to other aggregation algorithms. This demonstrates that the proposed framework has a minimal impact on the FL-based NTC training process.

For future work, we consider creating a lightweight implementation of the SHeRAA-FL framework to ensure it has a minimal processing impact on resource-constrained edge devices. We also plan to expand the evaluation to other application areas and with a larger number of edge clients. Moreover, we plan to integrate mechanisms that properly handle non-IID and imbalanced datasets from FL clients to enhance the global NTC model's performance. Lastly, we plan to leverage explainable AI techniques to better understand the decisions the global NTC model makes and the contribution of each client's local model to the overall performance.

## ACKNOWLEDGEMENTS

The authors would like to acknowledge Professor Asokan from University of Waterloo for his guidance and reviewing this work.

### Funding

The work was supported by the Fundamental Research Grant Scheme (FRGS/1/2023/ICT11/UM/02/1) awarded by the Ministry of Higher Education of Malaysia. The funders had no role in study design, data collection and analysis, decision to publish, or preparation of the manuscript.

### Grant Disclosures

The following grant information was disclosed by the authors:
Fundamental Research Grant Scheme: FRGS/1/2023/ICT11/UM/02/1.
Ministry of Higher Education of Malaysia.

### Competing Interests

The authors declare that they have no competing interests.

### Author Contributions

- Azizi Ariffin conceived and designed the experiments, performed the experiments, analyzed the data, performed the computation work, prepared figures and/or tables, authored or reviewed drafts of the article, and approved the final draft.
- Faiz Zaki conceived and designed the experiments, analyzed the data, authored or reviewed drafts of the article, and approved the final draft.
- Hazim Hanif conceived and designed the experiments, analyzed the data, authored or reviewed drafts of the article, and approved the final draft.
- Nor Badrul Anuar conceived and designed the experiments, analyzed the data, authored or reviewed drafts of the article, and approved the final draft.

## Data Availability

Code is available at GitHub and Zenodo:

- https://github.com/mebikarbonat/SHeRAA-FL.

- Muhammad Azizi. (2025). mebikarbonat/SHeRAA-FL: SHeRAA-FL (v0.10). Zenodo. https://doi.org/10.5281/zenodo.15300137.

The ISCXVPN2016 dataset is available at https://www.unb.ca/cic/datasets/vpn.html. A request to download the ISCXVPN2016 Dataset must be submitted here: http://cicresearch.ca/CICDataset/ISCX-VPN-NonVPN-2016.

Fashion-MNIST Dataset is available at GitHub: https://github.com/zalandoresearch/fashion-mnist

CIFAR-10 Dataset is available at: https://www.cs.toronto.edu/~kriz/cifar.html

N-BaIoT Dataset is available at: Meidan, Y., Bohadana, M., Mathov, Y., Mirsky, Y., Breitenbacher, D., , A., & Shabtai, A. (2018). detection_of_IoT_botnet_attacks_N_BaIoT [Dataset]. UCI Machine Learning Repository. https://doi.org/10.24432/C5RC8J.

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
