# Peer review of "Mitigating adversarial attacks in federated learning based network traffic classification applications using secure hierarchical remote attestation and adaptive aggregation framework"

_PeerJ Computer Science, doi:10.7717/peerj-cs.3215_

## Round 0.1 · original submission · Major Revisions

**Language Note:** The review process has identified that the English language must be improved. PeerJ can provide language editing services - please contact us at [email protected] for pricing (be sure to provide your manuscript number and title). Alternatively, you should make your own arrangements to improve the language quality and provide details in your response letter. – PeerJ Staff

·

Basic reporting

1. Abstract: The background should be summarized in a maximum of four lines, and the problems should be clearly stated. The whole abstract, including background, issues, methods, and results, should be summarized to a maximum of 250 words.

2. Introduction: This section requires serious author attention by restructuring the background of the study, defining some of the important terms, such as Network traffic, adversarial attacks, and federated learning. An in-depth discussion about those terms, causes, and other related information should be provided.

3. This discussion in lines 124-143 is inappropriate in section 1; move it to the results and discussion section.

4. Related Works: All the subheadings in this section should be deleted and merged as related works. However, recent and related articles between 2020-2025 should be reviewed, and also present the summary in a meta-analysis table to show the research trends and gaps by including adversarial attack types, methods, strengths, limitations, and application area.

5. * The adversarial attacks and defensive approach should be discussed in the Background section.

Experimental design

7. The FL-Based NTC Architecture section in line 291 should be taken as a methodology for discussion and presentation.

8. Secure Framework Design in line 399 should be discussed in the methodology section or as a subsection in the methodology.

Validity of the findings

9. Evaluation Metrics in line 656 should be moved to the results and discussion section

10. The accuracy and F1 as performance evaluation metrics should be well established and justified.

11. How many traffic categories does each adversarial attack have in the dataset? What is the size and classes? Please state clearly in your discussion how you handled the data imbalance, which can affect the minority accuracy in the training module. Use SMOTE or weighted loss functions to improve minority class performance.

12. Also, use SHAP or LIME to interpret model predictions with graphical analysis.

13. However, the title claims, “Mitigation Adversarial Attacks in Federated Learning.” How did you identify unusual patterns or behaviors that might indicate an adversarial attack, and what are the prevention measures provided?

14. * The results obtained should be evaluated and compared with the existing work in the literature

Additional comments

References: This section requires serious author attention to ensure it conforms to the journal reference style.

The article must be subjected to correction for typos and grammatical errors.

Each figure and table should be duly mentioned in the text before presentation and discussed.

Reviewer 2 ·

Basic reporting

The manuscript is well-structured. The abstract and introduction effectively summarise the motivation, background, and objectives. Figures and tables are relevant, clearly labelled, and support the narrative. The methodology is generally described in sufficient detail. However, the related work section would benefit from a deeper engagement with prior research on TPMs and TEEs in federated learning to better contextualise the contribution. Some variables are introduced without definitions; a concise notation table would enhance clarity. Additionally, a discussion of computational overhead is missing and would strengthen the paper’s practical framing.

Experimental design

The study presents original research within the journal’s scope, addressing a well-defined and meaningful question: how to secure federated learning-based network traffic classification against adversarial attacks. The proposed SHeRAA-FL framework combines TPM-based remote attestation, hierarchical training, and adaptive aggregation, a novel integration. The experiments are generally well-designed and simulate realistic adversarial scenarios using the ISCX-VPN 2016 dataset. Comparative evaluations with standard aggregation techniques add credibility. However, the experimental design omits an analysis of runtime and computational overhead, which is essential given the framework’s complexity. Additionally, although the methods are described in detail, the absence of statistical validation limits reproducibility under varying conditions.

Validity of the findings

The findings are generally valid and well-aligned with the research objectives. The conclusions are clearly stated and supported by the results presented. The study includes comparisons across multiple adversarial scenarios and aggregation strategies, which strengthens the credibility of the performance claims. However, the absence of statistical measures (e.g., variance, confidence intervals) limits the robustness of the results.

---

## Round 0.2 · accepted · Accept

The reviewer is satisfied with the authors' changes, and so I can recommend this article for acceptance.

Reviewer 2 ·

Basic reporting

The manuscript has significantly improved in proofreading and now follows a clear, logical structure with all standard sections included. The introduction defines key terms and offers background supported by recent, relevant literature (2020–2025), integrated into a cohesive Related Works section with a useful summary table highlighting research trends and gaps. Figures and tables are well-labelled and directly support the text. The article includes publicly available datasets, ensures transparency and reproducibility, and the results fully address the stated hypotheses, forming a coherent, publishable unit.

Experimental design

The study presents research that falls well within the journal’s aims and scope. The research question is clearly stated, addressing the critical gap of defending FL-based NTC systems against multiple types of adversarial attacks, including scenarios with multiple attackers. The experimental setup uses multiple public datasets (ISCX-VPN 2016, Fashion-MNIST, N-BaIoT, CIFAR-10) and evaluates the proposed framework against established baselines under both normal and adversarial conditions. The investigation is rigorous, technically sound, and adheres to ethical standards.

Validity of the findings

The experimental results are well-controlled, with appropriate baselines for comparison, and statistical measures are reported to support performance assertions. The conclusions are clearly linked to the research question and remain within the scope of the presented results, avoiding overstatement. The authors demonstrate that the proposed framework maintains high accuracy and resilience across various attack types and attacker scenarios, outperforming established defensive methods. All supporting data have been provided in accordance with the journal’s data sharing policy, ensuring transparency and reproducibility.